# Systematic comparison of tools used for m⁶A mapping from nanopore direct RNA sequencing

Zhen-Dong Zhong[1,4], Ying-Yuan Xie[1,4], Hong-Xuan Chen[1], Ye-Lin Lan[1], Xue-Hong Liu[1], Jing-Yun Ji[1], Fu Wu[1], Lingmei Jin [2], Jiekai Chen [2], Daniel W. Mak[3], Zhang Zhang[1] ✉ & Guan-Zheng Luo [1] ✉

N6-methyladenosine (m6A) has been increasingly recognized as a new and important regulator of gene expression. To date, transcriptome-wide m6A detection primarily relies on well-established methods using next-generation sequencing (NGS) platform. However, direct RNA sequencing (DRS) using the Oxford Nanopore Technologies (ONT) platform has recently emerged as a promising alternative method to study m6A. While multiple computational tools are being developed to facilitate the direct detection of nucleotide modifications, little is known about the capabilities and limitations of these tools. Here, we systematically compare ten tools used for mapping m6A from ONT DRS data. We find that most tools present a trade-off between precision and recall, and integrating results from multiple tools greatly improve performance. Using a negative control could improve precision by subtracting certain intrinsic bias. We also observed variation in detection capabilities and quantitative information among motifs, and identified sequencing depth and m6A stoichiometry as potential factors affecting performance. Our study provides insight into the computational tools currently used for mapping m6A based on ONT DRS data and highlights the potential for further improving these tools, which may serve as the basis for future research.

Over 150 types of RNA modifications have been discovered across all domains of life[1]. The most abundant and best-characterized type of modification found in the eukaryotic mRNA is N6-methyladenosine (m6A)[2,3]. The discovery of the m6A demethylase fat mass- and obesity-associated protein (FTO) highlights the reversible and dynamic nature of this RNA modification, which may serve as a novel regulator in gene expression[4]. Subsequent studies have revealed that m6A is modulated by a complex network composed of "writer", "eraser" and "reader" proteins[3], through which it can co-transcriptionally and post-transcriptionally influence essential RNA metabolic processes, including pre-mRNA splicing and polyadenylation as well as mRNA export, decay and translation[5]. Recent functional studies have demonstrated the essential role of m6A in physiological and pathological processes in various organisms, further advancing our knowledge and understanding of RNA biology[6,7].

Owing to the rapid development of m6A detection methods that have been developed on the next-generation sequencing (NGS) platform, the m6A landscape has since been documented in a number of

[1]MOE Key Laboratory of Gene Function and Regulation, Guangdong Province Key Laboratory of Pharmaceutical Functional Genes, State Key Laboratory of Biocontrol, School of Life Sciences, Sun Yat-sen University, 510275 Guangzhou, China. [2]CAS Key Laboratory of Regenerative Biology, Guangzhou Institutes of Biomedicine and Health, Chinese Academy of Sciences, Guangzhou, China. [3]School of Biomedical Sciences, LKS Faculty of Medicine, The University of Hong Kong, Hong Kong, China. [4]These authors contributed equally: Zhen-Dong Zhong, Ying-Yuan Xie. ✉e-mail: zhangzhang@mail.sysu.edu.cn; luogzh5@mail.sysu.edu.cn

different organisms[8]. Currently, there are two main NGS-based strategies that are employed in transcriptome-wide m6A mapping: (i) antibody-dependent methods, such as MeRIP-seq and miCLIP[9,10], and (ii) antibody-independent methods, such as m6A-REF-seq/MAZTER-seq[11,12] and DART-seq[13]. The most widely used MeRIP-seq protocol employs antibodies to enrich RNA fragments containing m6A; however, this approach suffers from the drawback of high false-positive rates as a result of non-specific antibody binding[14–16] and hence, inaccurate stoichiometric estimates. Meanwhile, m6A-REF-seq/MAZTER-seq utilizes an endonuclease that explicitly cleaves unmethylated ACA motifs but leaves the (m6A)CA intact. This approach precisely identifies m6A sites with single-base resolution and accurately quantifies methylation rates, yet it is limited to a specific sequence context. For example, while the endonuclease MazF specifically recognizes ACA motifs, it only covers approximately 16-25% of m6A sites of the whole transcriptome[11]. In DART-seq, the efficiency and specificity of exogenously expressed APOBEC1-YTH fusion proteins, which are known to induce C-to-U deamination at sites adjacent to m6A methylated sites, have not yet been thoroughly characterized. Most importantly, all of these NGS-based methods are limited to the detection of aggregate RNA molecule information and are incapable in accurately quantifying the stoichiometry at the molecular level. Accordingly, a novel m6A detection method with both high resolution and quantification capability is greatly needed to overcome these limitations.

In the last decade, third-generation sequencing (TGS) technologies represented by two platforms, namely Pacific Biosciences (PacBio) and Oxford Nanopore Technologies (ONT), have emerged as promising alternatives for mapping nucleotide modifications[17]. PacBio sequencing identifies DNA modifications using the interpulse duration (IPD), an approach requiring high sequencing coverage but suffers from low discriminatory power[18]. On the other hand, the ONT sequencing platform directly sequences DNA or RNA based on the principle of monitoring shifts in electric current caused by the traversing of a single molecule across a nanopore embedded within a synthetic polymer membrane. The effect on the flow of the current through the pore by modified bases may be distinct from their unmodified counterpart, thereby allowing for the detection of nucleotide modifications based on the identified electric current differentials[19,20].

A pioneering study published in 2018 reported for the first time the direct sequencing of native RNA molecules using the ONT platform[20]. Since then, at least ten computational tools have been developed to map the location of m6A methylation and to determine its stoichiometry from direct RNA sequencing (DRS) (Supplementary Fig. 1a and Supplementary Table 1). These tools can generally be divided into two classes based on the strategies employed to identify modified nucleotides (Fig. 1a). The first class relies on the detection of electric current differentials. Briefly, a continuous current is subdivided into "events" and then assigned to each nucleotide in a reference sequence using one of the two widely-used algorithms, namely Nanopolish-eventalign[19] or Tombo-resquiggle[21]. For each nucleotide, current features such as the median, mean, standard deviation and dwell-time (i.e., the time a 5-mer sequence resided in the nanopore) are extracted and used as inputs for one of three different classification methods: statistical testing (e.g., used in Tombo[21]), machine learning (e.g., used in MINES[22], Nanom6A[23] and m6Anet[24]) or clustering (e.g., used in Nanocompore[25] and xPore[26]) (Fig. 1a). The second class of tools utilizes base-calling "errors" resulting from the change in current caused by the presence of RNA modifications. Base-calling "errors" may represent mismatches, insertions, deletions or variability of base quality scores. These types of information are collected and classified using alignment results. To identify modified bases among the "errors", Epinano[27] uses a pre-trained machine learning model, while other tools (e.g., DiffErr[28], DRUMMER[29] and ELIGOS[30]) perform statistical testing to compare "errors" with an internal model or control sample. Given the high false-positive rate in defining "errors", a control sample with no (or low levels of) RNA modification is usually necessary.

As ONT DRS sequencing captures both isoform and nucleotide modification information, it has been successfully applied in dissecting

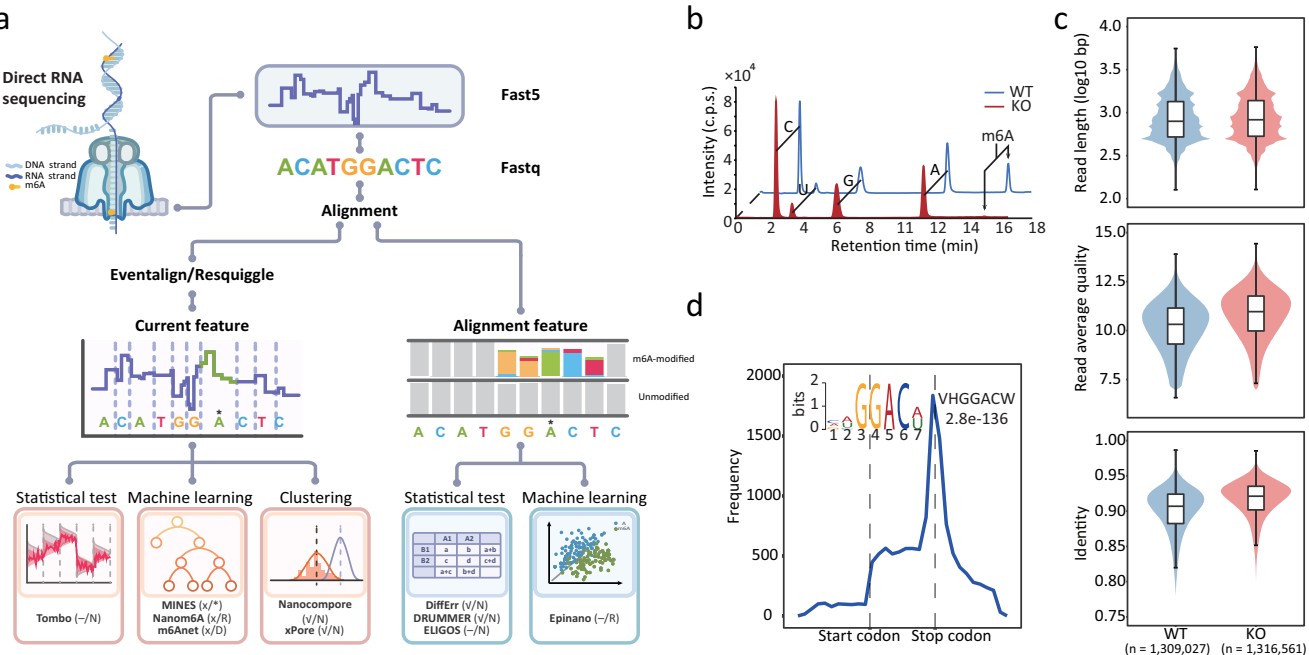

**Fig. 1 | Tools and datasets used for m6A mapping based on the ONT DRS platform. a** Pipeline and strategies employed by various m6A mapping tools. Information on the requirement for a negative control, minimum coverage and motif specificity for each tool is shown in brackets. "√" and "x" indicate the requirement for a negative control, while "-" indicates that a negative control is optional. "N" indicates a lack of motif limitations, while "D" and "R" indicate DRACH and RRACH, respectively. Note that MINES detects m6A in AGACT and GGACH motifs. **b** m6A detection and quantification by LC-MS/MS using WT and *Mettl3* KO samples. **c** Violin plots of ONT data features for WT and KO samples, the upper and lower limits represent the 75th and 25th percentiles, respectively, while the center line represents the median; upper and lower whiskers indicate ±1.5× the interquartile range (IQR). The number of reads is indicated in parentheses. **d** Metagene distribution plot with motif preferences of m6A peaks identified from the MeRIP-seq data.

the role of m6A in shaping transcriptome complexity, such as through alternative splicing[29,31], alternative polyadenylation[23], circular RNA biogenesis[32] and other transcriptional events[33]. Despite a number of highly sophisticated state-of-the-art computational tools that have been developed to detect and quantify m6A, a thorough evaluation and comparison of these tools is currently lacking. To address this gap, we compared ten computational tools used for m6A mapping from ONT DRS data and quantitatively evaluated their performance based on two continuous evaluation metrics (Receiver Operating Characteristic (ROC) and Precision Recall (PR) curves), followed by the identification of m6A under a cutoff that is associated with the best F1 score. We found that most tools presented a trade-off between precision and recall, and their performance largely improved with the integration of the results from multiple tools. We also assessed the intrinsic preference for m6A-irrelevant sites and demonstrated that introduction of a m6A-free negative control improved the performances of most tools. In addition, we found that the detection capability varied among motifs, as current differentials were less easily detected in certain sequence contexts. For the quantitative tools, a wide discrepancy was observed among their stoichiometric estimates, albeit with acceptable results in some of the motifs. We showed that this comprehensive comparison between the multiple tools serves as a guide for best practices in m6A mapping and quantification, as well as conducting of further analysis on the ONT DRS platform.

## Results

### Datasets used for ONT tool comparisons

In order to compare the computational tools currently available for m6A detection (Fig. 1a, Supplementary Fig. 1a, and Supplementary Table 1), we conducted ONT DRS for mRNA from wild-type (WT) mouse embryonic stem cells (mESCs) and corresponding m6A-deficient samples with the major m6A methyltransferase *Mettl3* knockout (KO). The mRNA derived from *Mettl3* KO samples were confirmed to be virtually absent of m6A by liquid chromatography-mass spectrometry (LC-MS/MS) (Fig. 1b). We obtained 1.31 and 1.32 million reads for the WT and KO samples, respectively, with similar median read length of 1 kb (Fig. 1c). The average read quality scores were greater than 10 and the median identities of aligned reads were 90.7% and 92.1% for WT and KO samples, respectively (Fig. 1c). The average quality and alignment identity of reads from WT samples were slightly lower than that from KO samples, suggesting that the presence of m6A may cause electric current variability, and thus lead to more base-calling "errors"[27] (Fig. 1c). We also calculated the coverage of aligned reads to annotated transcripts and found that a large proportion of reads were almost full-length in both samples (Supplementary Fig. 1b). Overall, the quality of the DRS data was sufficient for m6A detection.

To obtain validation datasets, we performed the well-characterized MeRIP-seq assay on WT and *Mettl3* KO samples. Peaks identified in WT samples were distributed around stop codons with a consensus sequence RRACH (R = A/G, H = A/T/C), while peaks from KO samples were randomly distributed (Supplementary Fig. 1c). Given the absence of m6A methylation in the KO samples, these m6A-irrelevant peaks were mainly due to the non-specific antibody binding. After excluding the m6A-irrelevant peaks, a total of 18,069 high-confidence m6A peaks were identified (Fig. 1d). The ONT DRS platform detected 903,637 and 895,031 RRACH motifs in WT and KO samples, respectively, and more than 40% (370,000) were supported by at least five reads (Supplementary Fig. 1d). The m6A sites identified in the same cell line by three independent NGS-based methods, i.e., m6A-REF-seq[12], miCLIP[34] and miCLIP2[35], were also included as validation sets. To further validate our results, two published DRS datasets derived from WT and *Mettl3* KO mESCs samples were included as replicates[30] (Supplementary Table 2). In addition, we also included published DRS datasets from WT *Arabidopsis* (Col-0) and mutants defective of VIRILIZER (*vir-1*) as validation in distant species[28] (Supplementary Table 2).

### A comparison between the performance of ten computational tools

In order to assess the aforementioned ten tools' capability to identify m6A sites, we evaluated their performance based on m6A sites identified using MeRIP-seq, miCLIP and miCLIP2 assays as ground truth (see Methods). In considering the limitations of each NGS-based method, we took the m6A sites identified by each NGS-based method or their intersection and union as validation sets separately. Since some tools are limited to RRACH motifs, we only retained the sites found in this specific sequence context (Supplementary Table 3). We then applied each computational tool to predict m6A and ranked the sites according to the confidence scores reported by each tool. As some tools require (DiffErr, DRUMMER, xPore, Nanocompore) or support (ELIGOS_diff, Epinano_delta, Tombo_com) to use negative control sample, while others (ELIGOS, Epinano, Tombo, m6Anet, MINES and Nanom6A) do not, we therefore evaluated their performances in two separate modes which are hereinafter referred to as the compare-mode and single-mode.

We first quantified their performance using two continuous evaluation metrics, the area under the ROC curve (ROC AUC) and PR curve (PR AUC). Among the tools using single-mode, m6Anet outperformed others by achieving a ROC AUC of 0.68-0.94 and PR AUC of 0.20-0.52, followed by ELIGOS, Epinano and Nanom6A, while Tombo achieved the lowest ROC AUC and PR AUC (Fig. 2a, b and Supplementary Fig. 2a). As MINES trained its models on 12 RRACH motifs but filtered the output m6A sites except that located on four sequences (GGACT, GGACA, GGACC, AGACT), we modified its code to output all sites. Even though MINES achieved the second lowest ROC AUC, the PR AUC was relatively high (Fig. 2a, b and Supplementary Fig. 2a). Notably, all tools performed much better on four motifs (GGACT, GGACA, GGACC, AGACT), though their ranking remained unchanged (Supplementary Fig. 3). In the compare-mode, xPore achieved the best ROC AUC of 0.75-0.91 and PR AUC of 0.27-0.67, which were slightly higher than those of Epinano_delta. Three error-based tools, ELIGOS_diff, DRUMMER and DiffErr, achieved almost the same performance, which may be due to the similar principle they follow (Fig. 2c, d and Supplementary Fig. 2b). When using different validation sets, the ranking of each tool remained consistent, indicating that the relative performance of the tools is not affected by the choice of validation set (Fig. 2a–d and Supplementary Fig. 2a, b). In addition, overall ranking was not changed though the ROC and PR AUC of all tools improved by taking the maximum of the predicted probability within each peak from MeRIP-seq results (see Methods) (Supplementary Fig. 2c).

We next assessed the precision of the top 20,000 m6A sites predicted by each tool. Among the tools in single-mode, m6Anet constantly showed the highest precision, with 90% of the top 2000 sites or 80% of the top 5000 sites being supported by at least one NGS-based method (Fig. 2e, f and Supplementary Fig. 2d, e). In the compare-mode, xPore achieved the best precision for top 4000 m6A sites, but its performance decreased sharply with the inclusion of additional sites. In contrast, Epinano_delta constantly showed the best precision for top 4000 to 20,000 m6A sites (Fig. 2g, h and Supplementary Fig. 2f, g). When analyzing the relative location of top m6A sites, we found that those with higher precision showed greater enrichment around stop codons, a characteristic distribution of m6A (Fig. 2i, j). This was consistent with previous findings in mammalian species[9,10]. The presence of the GGACU consensus sequence was more pronounced in m6A sites identified by tools with higher precision, also supporting previous observations about the distribution of m6A (Fig. 2i, j).

To further confirm the evaluation of each tool, we repeated the aforementioned process on a published ONT DRS dataset that was derived from WT and *Mettl3* KO mESCs samples[30]. We found that all single-mode tools achieved nearly identical performance to their performance using our data; tools in the compare-mode, on the other hand, generally performed worse, though their ranking remained

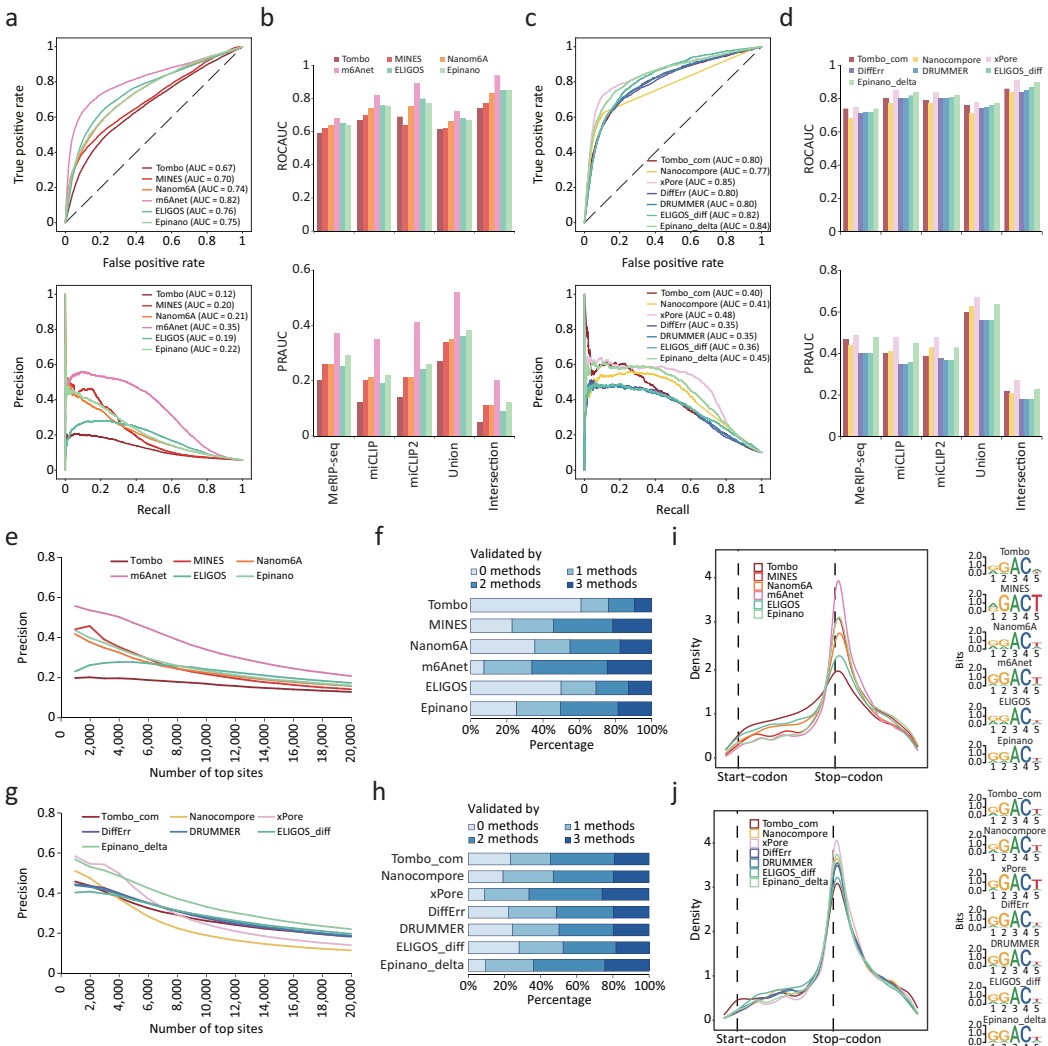

**Fig. 2 | Performance evaluation of ten tools for m6A detection capability.**
**a** Receiver operating characteristic (ROC) curve (top) and Precision Recall (PR) curve (bottom) for candidate sites detected by single-mode tools using the miCLIP data as ground truth. Area under the curve (AUC) is indicated in parentheses. **b** ROC AUC (top) and PR AUC (bottom) for candidate sites detected by single-mode tools using different NGS-based datasets as ground truth. **c** ROC curve (top) and PR curve (bottom) for candidate sites detected by compare-mode tools using the miCLIP data as ground truth. AUC is indicated in parentheses. **d** ROC AUC (top) and PR AUC (bottom) for candidate sites detected by compare-mode tools using different NGS-based datasets as ground truth. **e** Precision curve for top 20,000 m6A sites detected by single-mode tools using the miCLIP data as ground truth. **f** Percentage

for m6A sites supported by various number of NGS-based methods. Top 2000 m6A sites detected by single-mode tools are counted. **g** Precision curve for top 20,000 m6A sites detected by compare-mode tools using the miCLIP data as ground truth. **h** Percentage for m6A sites supported by various number of NGS-based methods. Top 2000 m6A sites detected by compare-mode tools are counted. **i** Metagene plots (left) of the transcriptome-wide m6A distribution for top 2000 m6A sites detected by single-mode tools. The sequence motifs of these sites plotted by Seqlogo are shown on the right. **j** Metagene plots (left) of the transcriptome-wide m6A distribution for top 2000 m6A sites detected by compare-mode tools. The sequence motifs of these sites plotted by Seqlogo are shown on the right.

unchanged (Supplementary Fig. 4a–d). Notably, the published mESCs was found to be an incomplete *Mettl3* KO cell line with an estimated ~40% of m6A still present[30,36]. This implies that a perfect negative control absent of any m6A would be crucial for these compare-mode tools. Likewise, these tools showed superior m6A detection capability in the aforementioned motifs (GGACT, GGACA, GGACC, AGACT) (Supplementary Fig. 4c, d).

As some tools were constructed and tested on training data from mammalian species, they may have different performance in non-mammalian species in which the sequence preference and stoichiometry of m6A may vary. To further compare these tools in non-mammalian species, we evaluated their performance using published DRS data from the *Arabidopsis*[28]. As expected, the performance of machine-learning based tools was compromised when applied to the *Arabidopsis* dataset, particularly m6Anet and MINES which were

trained on DRS data from human samples using miCLIP as the ground truth (Supplementary Fig. 5). We reasoned that these two tools are not trained extensively on AAACH motifs, which are the predominantly occurring m6A context in *Arabidopsis* rather than AGACT and GGACH motifs in human or mouse (Supplementary Table 4). On the other hand, most error-based tools performed better in *Arabidopsis* (Supplementary Fig. 5), likely benefiting from a higher coverage (Supplementary Table 3). Accordingly, given that the size of *Arabidopsis* genome is only 1/20 compared to that of the mouse genome, a higher coverage was achieved with similar sequencing reads (Supplementary Table 3), leading to more base-calling "errors" occurred in the presence of m6A and increasing the statistical power of error-based tools. However, as more than 10% of m6A were retained in *vir-1* samples[28], the performance of the compare-mode did not significantly improve (Supplementary Fig. 5). Similarly, the performance of the tools

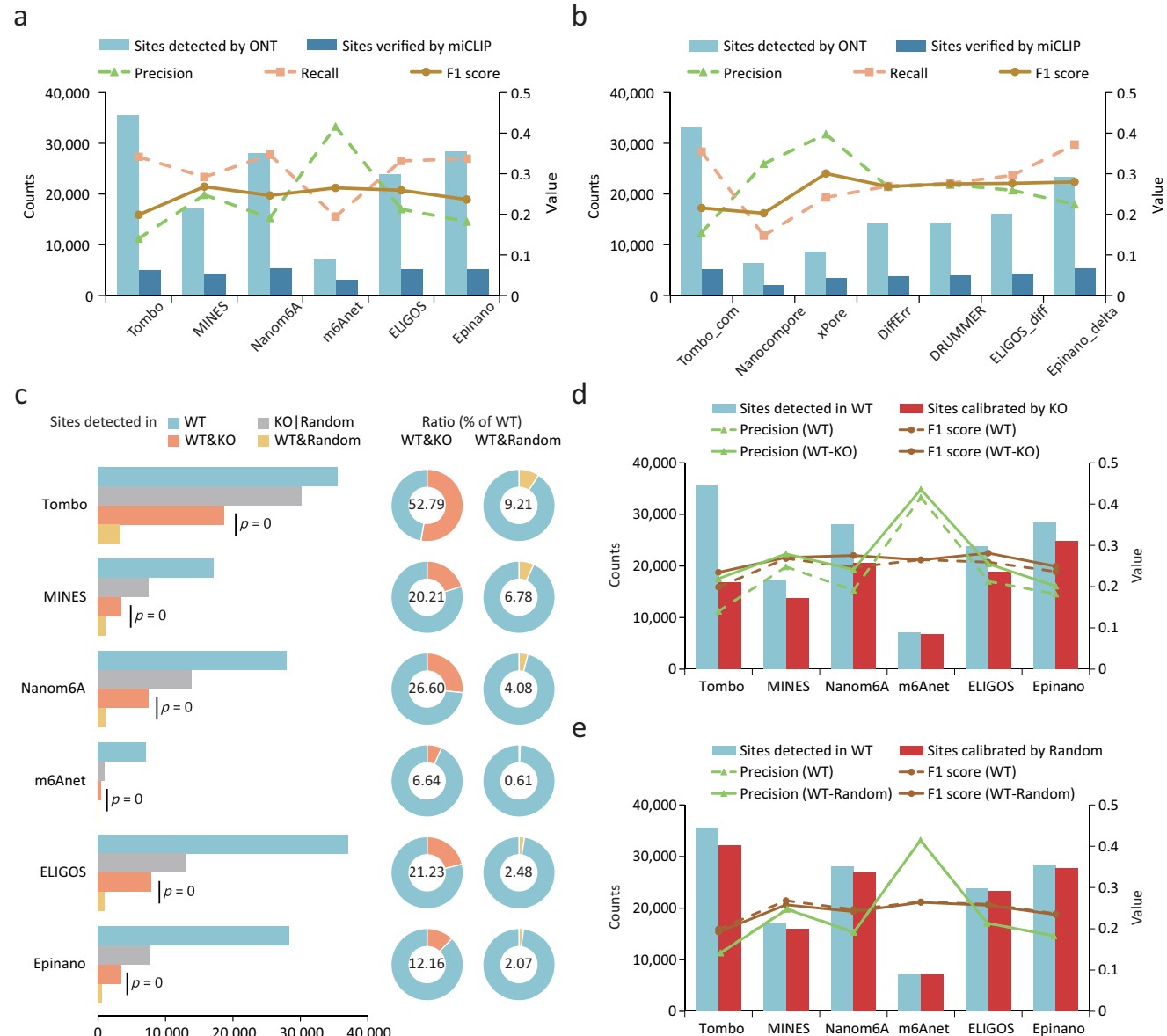

**Fig. 3 | Assessment of intrinsic bias and the effects of using a negative control on m6A detection. a, b** Precision, recall and F1 scores for single-mode tools (**a**) and compare-mode tools (**b**) using miCLIP results as validation set. **c** Counts and percentages of recurrent m6A sites identified amongst samples. One-sided Hypergeometric tests were applied to calculate significance. WT&KO, both in WT and KO; WT&Random, both in WT and Random; KO|Random, the total number in KO or Random. **d, e** The m6A site counts and precision and F1 scores before and after calibration with the results from KO (**d**) or random (**e**) samples. miCLIP results were used as a validation set. WT-KO, precision and F1 score for sites calibrated using KO results. WT-random, precision and F1 score for sites calibrated with a randomly selected dataset.

improved in four motifs (GGACT, GGACA, GGACC, AGACT), particularly for m6Anet and MINES (Supplementary Fig. 6). We observed exactly the same results from other replicate dataset from *Arabidopsis* (Supplementary Fig. 7).

In summary, we found that the performance of computational tools for detecting m6A sites varied significantly. Among those that only used a wild-type (WT) sample, m6Anet performed the best. In contrast, tools that used both WT and m6A-deficient (KO) samples exhibited a comparable performance, with the exception of xPore and Nanocompore, which omitted a number of m6A sites to be tested but had higher precision within the top sites. Tombo, ELIGOS and Epinano all had improved performance when using their compare-mode with a nearly m6A-free sample as negative control. In addition, machine-learning based tools may be affected by species-specific differences in m6A distribution and stoichiometry.

### Intrinsic bias of ONT tools in m6A detection

To determine the optimal cut-off value for each tool, we calculated the F1 scores under continuous cut-offs and selected the value that corresponded to the summed maximum F1 score when applied to the validation sets from miCLIP and miCLIP2 (Supplementary Fig. 8 and Supplementary Table 5). We detected 6410 to 35,475 m6A sites from WT samples and compared them to the validation sets from miCLIP and miCLIP2 (Fig. 3a, b and Supplementary Fig. 9a, b). Among the single-mode tools, Tombo, Nanom6A, ELIGOS and Epinano detected more m6A sites with higher recall rates, but at the cost of lower precision (Fig. 3a). While m6Anet and MINES omitted many of the sites to be tested and lost recall rates due to the high coverage requirement or motifs limitation, they achieved the highest F1 score thanks to their high precision (Fig. 3a). Among the tools in the compare-mode, two tools based on clustering principle achieved the highest precision;

however, it was challenging to balance precision and recall rate for other tools, likely due to the failure of clustering (see Methods; Fig. 3b). DiffErr, DRUMMER and ELIGOS_diff achieved almost the same performances (Fig. 3b). Similar results were found when using the validation set from miCLIP2 (Supplementary Fig. 9a, b). We also calculated precision, recall and F1 score with higher coverage requirement. Increasing the coverage from 5 to 20 resulted in a better F1 score for all tools, with most of them achieving a better precision and some achieving a better recall. However, increasing the coverage from 20 to 50 did not result in better performance for most tools (and even slightly worse for some). Notably, the ranking of tools was nearly the same regardless of different coverage requirements (Supplementary Fig. 9). The performance of Tombo, ELIGOS and Epinano was better in compare-mode than in single-mode, particularly in terms of precision (Fig. 3a, b). This suggests that there might be an intrinsic preference for certain sites that is unrelated to m6A and can be avoided when using the compare-mode.

As the *Mettl3* KO cell we used were verified to be free of m6A (Fig. 1b), we expected no (or few) m6A sites would be detected using the single-mode tools. Strikingly, a substantial number of m6A sites (from 983 to 30,207) were identified in the KO sample by most tools, with Tombo even detecting a similar number of m6A sites in KO and WT samples (Fig. 3c). This suggests a systematic overestimation of m6A abundance by the tools. Sites detected in the KO sample did not show clear enrichment around the stop codons when compared to the background (Supplementary Fig. 9c), thus implying a prevalence of false positives in the results. We found that the recurrence of m6A sites in WT samples was significantly higher in KO samples than in random datasets (see Methods; Fig. 3c). For instance, 52.79% and 28.68% of the m6A sites detected in the WT samples were also detected in the KO samples by Tombo and Nanom6A, respectively, while only 9.21% and 4.52% of sites could be found in the random datasets (Fig. 3c), thus suggesting a potential bias towards certain m6A-irrelevant sites. We inspected these sites that were detected in both WT and KO samples and found that single-nucleotide polymorphisms (SNPs) and insertions/deletions (indels) near RRACH motifs could contribute to this bias, particularly for ELIGOS, which detects m6A by comparing "errors" to the background model and treat SNPs/indels as "errors" during m6A detection (Supplementary Fig. 9d, e). In addition, homopolymer near the RRACH motif was also one of the causes of the bias (Supplementary Fig. 9d, e), as they may lead to deletion "errors" signal and inaccurate events assignment. Despite these observations, a proportion of sites detected in both WT and KO samples could not be explained.

We next employed the KO samples as a negative control to test whether it could help to identify false positives and thereby refine the list of reported m6A sites. After subtracting the sites detected in the negative control, we found that the precision of most tools in the single-mode were significantly improved, especially for Tombo and Nanom6A (Fig. 3d and Supplementary Fig. 9h). Despite decreased recall rates, the F1 scores of these tools were improved (Fig. 3d and Supplementary Fig. 9f, h). In contrast, calibrating m6A sites using sites in random datasets mentioned above did not improve the precision, but decreased the recall rates (Fig. 3e and Supplementary Fig. 9g, i). ELIGOS and Epinano performed better when using the compare-mode than directly subtracting the sites in negative control sample (Supplementary Fig. 9j, k). Overall, the intrinsic bias of each tool may lead to pervasive false positives and using a negative control can improve their performance.

## Detection capabilities varied among motifs

Most tools train separate machine learning models or build unique reference models to detect m6A sites with different sequence contexts (or motifs). Notably, m6A sites in AGACT/GGACH motifs tend to have better evaluation metrics (Supplementary Fig. 3). We hypothesized that detection capabilities may vary among motifs. We found that the recall rates for certain motifs, such as GGACT, GGACA, GGACC and

AGACT, were significantly higher than that of other motifs. On the other hand, some motifs such as AAACC, AAACA and GAACA showed extremely low recall rates (Fig. 4a). In addition to the highest recall rates observed in the four motifs (GGACT, GGACA, GGACC and AGACT), they also consistently showed the highest precision (Fig. 4b), suggesting much better detection capabilities of most tools for these motifs. This finding was consistent for both current-based and error-based tools (Supplementary Table 6), implying a difficulty in differentiating or detecting m6A sites within some 5-mer motifs. Similar results were observed when using the miCLIP2 result as a validation set (Supplementary Fig. 10a, b and Supplementary Table 6).

Given these findings, we suspected that electric current differentials between the modified and unmodified bases may be modest for some motifs, resulting in difficulties of distinguishing m6A from the Adenosine base (A). To test this hypothesis, we inspected 239 GGACA and 59 GAACA sites identified by m6A-REF-seq that were almost fully methylated in WT samples but non-methylated in KO samples (methylation rate of WT - KO > 0.8). Next, we analyzed ONT reads covering these m6A sites (for both WT and KO samples) to examine discrepancies in currents and base-calling "errors" profiles. Notably, current intensity differed greatly between modified and unmodified GGACA motifs but not GAACA motifs (Fig. 4c). In addition, modified reads with GGACA motif resulted in more base-calling "errors" compared to unmodified reads, especially for deletions and mismatches; in contrast, modified reads with GAACA motif showed little effect on the frequency of base-calling "errors" (Fig. 4d, e and Supplementary Fig. 10c). Nevertheless, the qualities of modified reads with either GGACA or GAACA motif were significantly lower than that of unmodified reads (Supplementary Fig. 10c). We concluded that current differentials between m6A and A were relatively small at some sequence contexts, leading to poor performance of most tools encountering such motifs.

## Performance of m6A quantification

Among the ten computational tools evaluated, three of the current-based tools (Tombo/Tombo_com, Nanom6A and xPore) are able to quantify methylation rates. Tombo/Tombo_com and Nanom6A detect m6A signal of single reads and thus allow for quantification of m6A at each genomic location covered by multiple reads, while xPore estimates the methylation rate by modeling a mixture of two Gaussian distributions corresponding to unmodified and modified reads. To evaluate the ability of each tool to quantify m6A sites, we compared their results to those obtained from NGS-based methods. We first calculated the enrichment score of each site using MeRIP-seq data, then divided the sites into seven categories according to their enrichment scores. When comparing the methylation rates for sites belonging to different categories, we found that sites with high enrichment scores also showed higher methylation rates (Fig. 5a). For all three tools, methylation rates were positively correlated with the enrichment score, and xPore showed the highest correlation (Fig. 5a). To further assess their performance quantitatively, we utilized the validation set from m6A-REF-seq results containing quantitative information for individual m6A sites in four motifs. These three tools showed largely different correlations with the validation set, among which Tombo and xPore achieved the lowest and the highest correlations, respectively (Pearson' $r = 0.21$ for Tombo and Pearson' $r = 0.57$ for xPore) (Fig. 5b). Correspondingly, the results of different tools were poorly correlated, with the lowest correlation between Tombo and Nanom6A (Pearson' $r = 0.27$), and the highest correlation between Nanom6A and xPore (Pearson' $r = 0.65$) (Fig. 5b). In addition, Tombo_com always showed a higher correlation with other methods than Tombo.

Given that previous reports on variable detection capabilities among different motifs, we investigated whether the performance of m6A quantification also varied among different sequence contexts. Despite the low overall consistency among the three tested tools, we

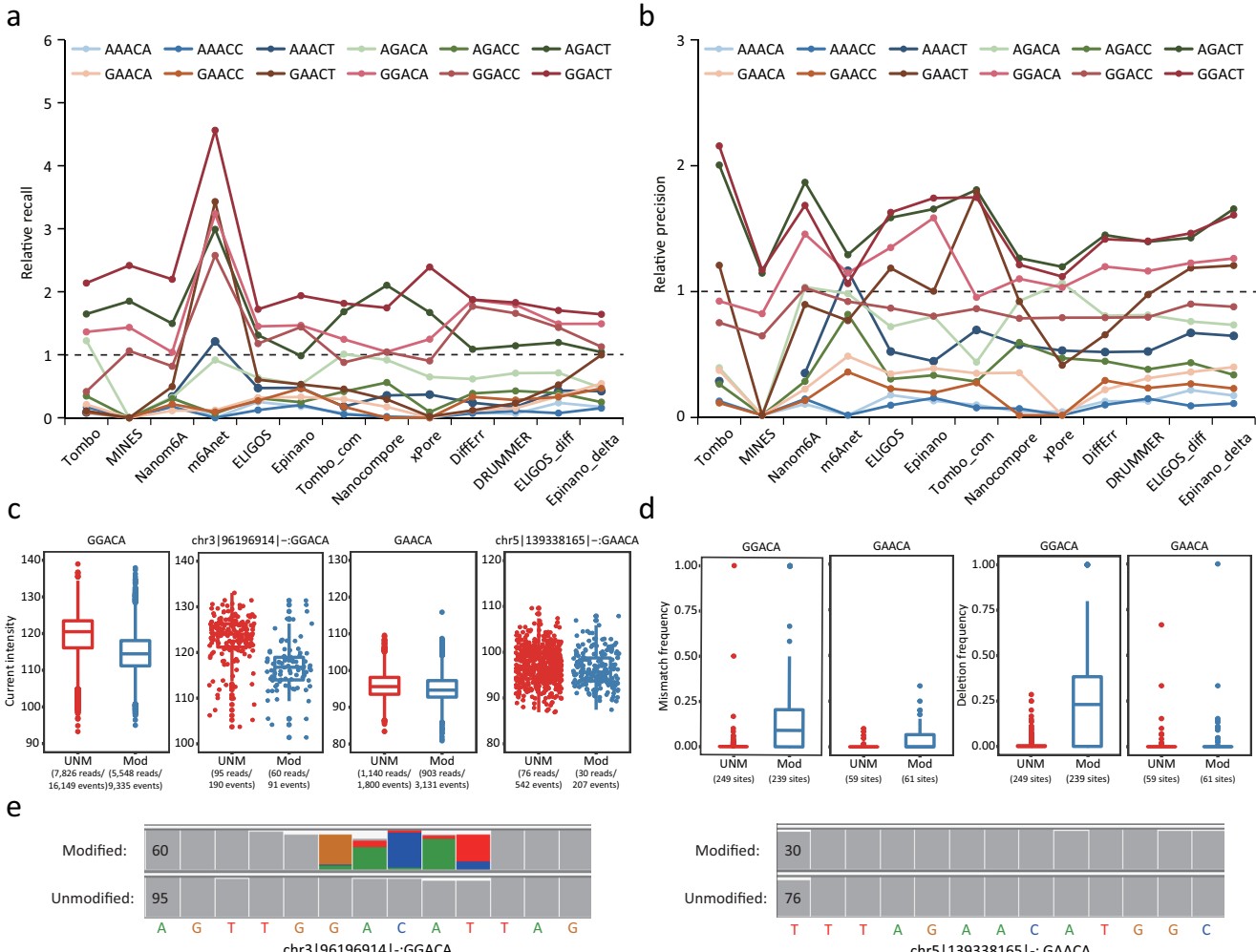

**Fig. 4 | Detection capabilities for m6A among sequence motifs. a, b** Relative recall rates (**a**) and precision (**b**) for 12 motifs bearing the consensus sequence RRACH. Relative recall rates/precision were calculated as the ratio of the recall/precision for the sites in a specific motif to the overall mean (across all sites). A m6A profile created from miCLIP results was used for validation. **c** Current intensity of modified and unmodified sites in GGACA or GAACA motifs. The m6A candidate sites (methylation rate of WT−KO > 0.8) were identified using m6A-REF-seq and marked as "Mod" (modified, sites in WT sample) or "UNM" (unmodified, sites in KO sample). This dataset was also used in **d, e**. For boxplots in **c, d**, the upper and lower

limits represent the 75th and 25th percentiles, respectively, while the center line represents the median; upper and lower whiskers indicate ±1.5× the interquartile range (IQR). The number of reads and events aligned to the sites are indicated in parentheses. **d** A comparison of mismatch and deletion frequencies for modified and unmodified sites in GGACA and GAACA motifs. The number of sites is indicated in parentheses. **e** IGV snapshots of two specific sites in different motifs (GGACA and GAACA). Modification status (modified or unmodified) was determined by m6A-REF-seq. Sites with allele frequency ≥0.1 are indicated with corresponding colors. Coverage for each site is indicated on the left of each short sequence.

found that they consistently showed higher correlations in specific motifs (Fig. 5c and Supplementary Fig. 11a–f). For instance, when comparing Tombo and Nanom6A, Pearson correlation coefficients for m6A sites within the GGACT and AGACT motifs increased to 0.7, a value significantly higher than that for all sites (0.27) (Fig. 5c). Similar results were observed for the other five pairs of comparisons (Supplementary Fig. 11a–e). Furthermore, increasing the coverage did not improve the quantification performance (Fig. 5d and Supplementary Fig. 11g).

## Detection capability is influenced by sequence coverage and m6A stoichiometry

In order to test whether the detection capability is affected by sequencing coverage, we compared the recall and precision across different coverage intervals for each tool. Three error-based tools (ELIGOS/ELIGOS_diff, DRUMMER and DiffErr) and Nanocompore were susceptible to low coverage, with their recall rates increasing dramatically as coverage increased (Fig. 6a), which explains the superior performance of these tools in *Arabidopsis* with high sequencing depth

(Supplementary Fig. 5). xPore and Tombo_com showed slightly lower recall rates at higher coverage interval, while other tools remained stable (Fig. 6a). In terms of precision, Tombo_com and Epinano_delta benefited most from the high coverage as did the performance of Epinano, Nanom6A and MINES, whereas other tools were less affected by the coverage (Fig. 6b). These results were consistent when using data from miCLIP2 as a positive control (Supplementary Fig. 12 and Supplementary Table 7). In general, the detection capabilities of different tools varied in their response to coverage (Fig. 6c and Supplementary Table 7).

As most m6A sites were partially methylated, we wondered whether the algorithms employed by different tools were also effective in detecting m6A sites with low methylation level. To test this, we divided m6A sites that were detected by m6A-REF-seq into five categories according to the methylation rate and compared the recall rate between different categories for each tool. As expected, it is difficult for all the tools to detect sites with low m6A levels (0-0.2); in contrast, the recall rate is largely improved for the sites with high m6A levels (0.8–1) (Fig. 6d and Supplementary Table 8). To further investigate all

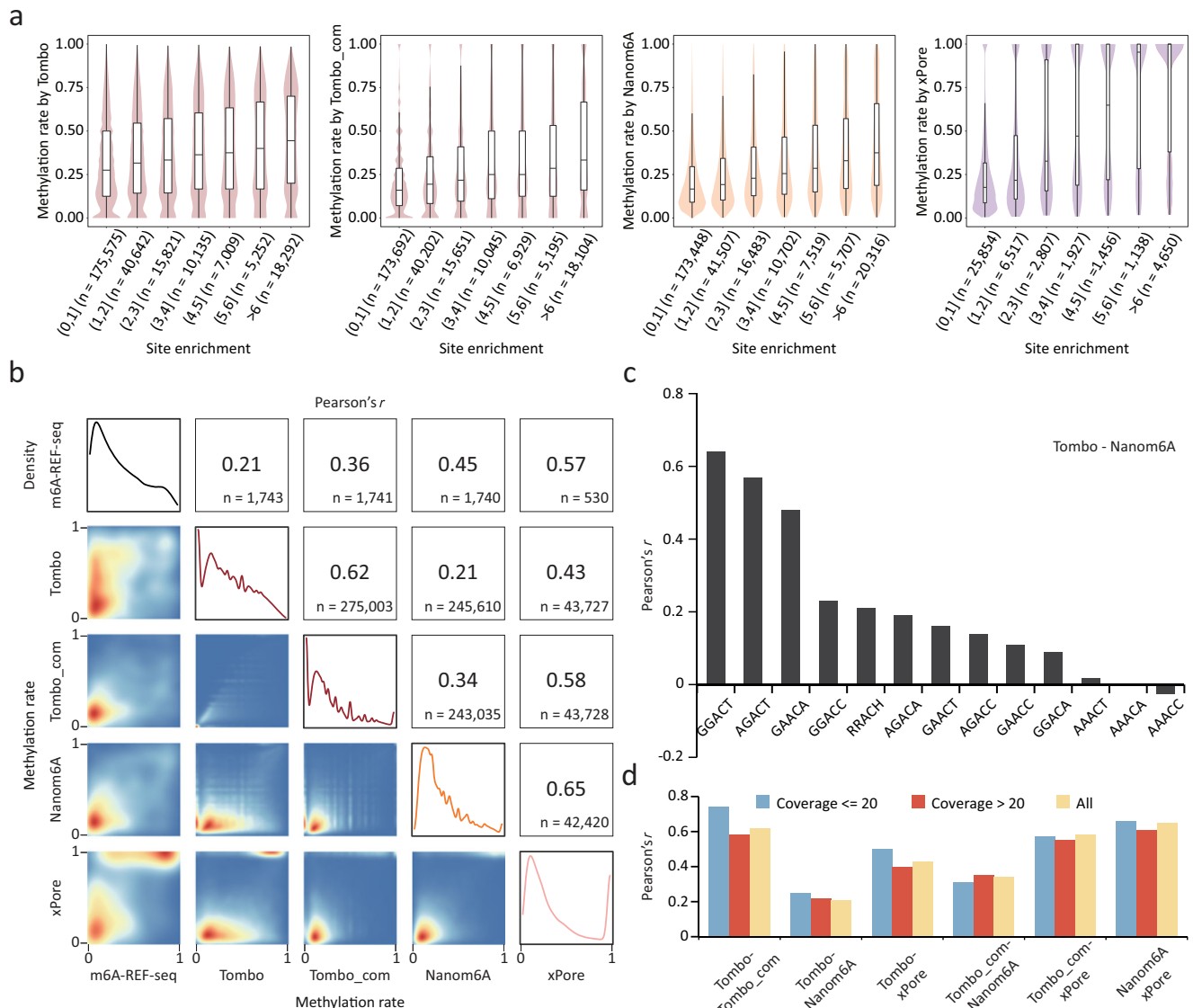

**Fig. 5 | Performance of Tombo/Tombo_com, Nanom6A and xPore for m6A quantification. a** The distribution of methylation rates determined by each of the three tools. m6A sites are classified into categories according to enrichment scores derived from MeRIP-seq. Violin plots show the distribution of detected methylation rates. In the boxplots, the lower and upper limits represent the 25th and 75th percentiles, respectively; the center line represents the median; and upper and lower whiskers indicate ±1.5×IQR. The number of sites occurring in each peak category are indicated in parentheses. **b** Correlations of methylation rates derived from m6A-REF-seq and those from each of the three tools. Pearson correlation coefficients and the number of sites for each paired comparison are indicated at the top-right, with 2D-density plots of the methylation rates at the lower-left. The four density plots on the diagonal show the distribution of methylation rates in the m6A-REF-seq results and those of each tool. **c** Correlation coefficients of methylation rates derived from Tombo and that from Nanom6A. m6A sites in different motifs are separately calculated. **d** Correlation coefficients of methylation rates derived from paired tools. m6A sites with different sequencing coverage are separately calculated.

RRACH motifs, we divided the m6A sites detected by miCLIP/miCLIP2 into six categories according to the site enrichment score quantified by MeRIP-seq. Similar results were observed for most tools, with some exception when using data from the miCLIP as positive control (Fig. 6e, f and Supplementary Table 8). In conclusion, m6A sites with higher methylation rates were easier to detect than those with lower methylation rates.

## Integrated analyses of multiple tools

As presented in the above, it can be difficult to balance precision and recall rates for most individual tools, even under their optimal cut-off settings. For example, Epinano_delta achieved the highest recall rate (0.37) but with a very low precision (0.22), whereas m6Anet achieved the best precision (0.42) at the cost of a low recall rate (0.19)

(Fig. 3a, b). To test whether a combination of multiple tools can improve performance, we intersected the top 5000 m6A sites reported by the different tools. Though the results of ELIGIS_diff, DRUMMER and DiffErr overlapped well with each other (0.6–0.8), wherein the overlap ratio was less than 0.5 for most pairs of other tools (Fig. 7a). A low overlap ratio may indicate many false-positives as reported by most tools; alternatively, different tools may capture different sets of authentic m6A sites and the integration of these results could improve the prediction performance. After merging the results reported by ten tools, we identified a total of 112,885 m6A sites. However, 54.98% (62,060) of the m6A sites were supported by only one tool, and only 6.66% (7523) of m6A sites were supported by more than half of the tools (Fig. 7b). Strikingly, only 251 (0.22%) of the m6A sites were identified by all of the tools (Fig. 7b). We then compared the precision,

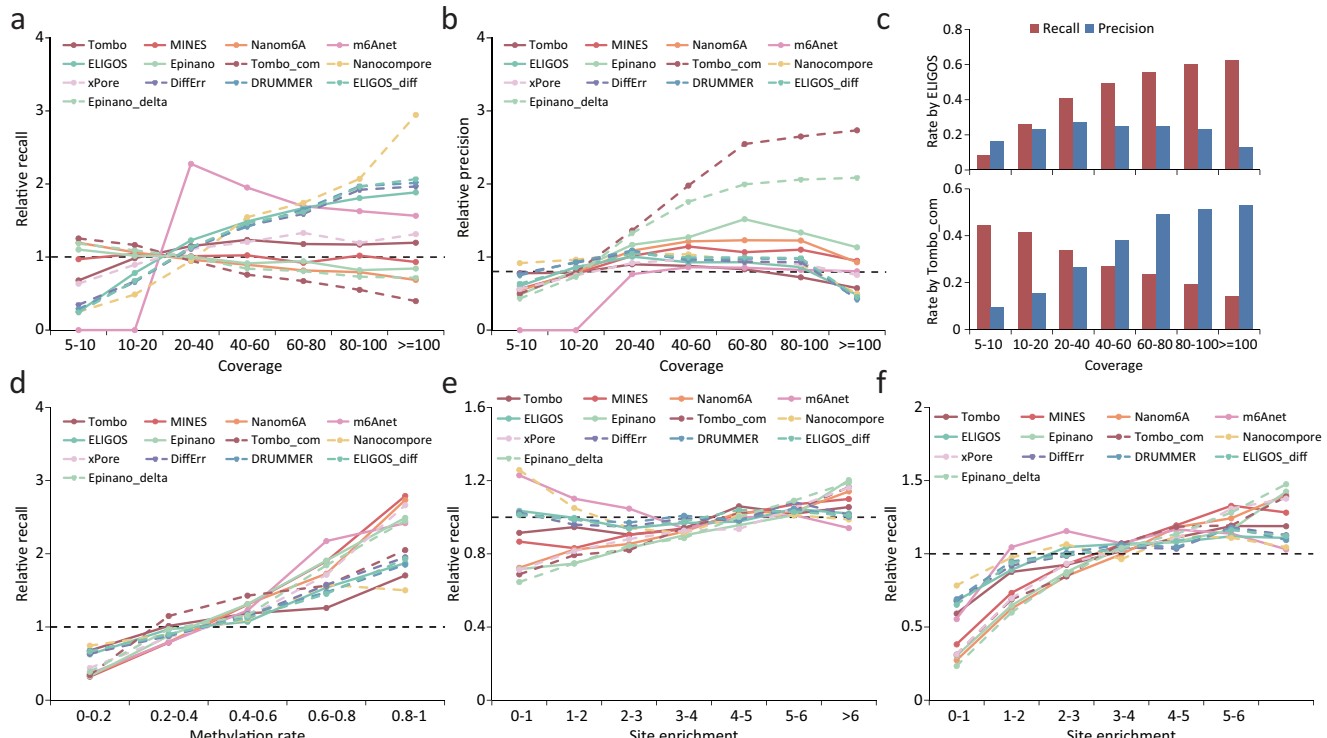

**Fig. 6 | Detection sensitivity for m6A sites varies under different coverage and stoichiometry. a, b** Relative recall rates (**a**) and precision (**b**) for m6A sites under different coverage. Relative recall rates/precision were calculated as the ratio of the recall rate/precision for m6A sites with a given coverage to the overall mean (across all sites). The validation set was derived from the miCLIP results. The same validation set is also used in **c. c** Recall rates and precision of ELIGOS and Tombo_com for m6A detection under different sequencing coverage. **d–f** Relative recall rates for m6A sites with different methylation rate/site enrichment. Relative recall rates were calculated as the ratio of the recall rate for m6A sites with a given methylation rate/site enrichment to the overall mean (across all sites). The validation sets were derived from the m6A-REF-seq (**d**), miCLIP (**e**) and miCLIP2 (**f**) results.

recall and F1 scores of the integrated predictions to that of the same number of top m6A sites from each individual tool and found that the integrated predictions achieved a better performance in terms of precision, recall rate and F1 score (Fig. 7c). Moreover, using m6A sites supported by more than 4 or 5 tools achieved the highest F1 score (Fig. 7c), thus highlighting the effectiveness of integrating the predictions from multiple tools.

Considering that TGS-based methods or NGS-based methods for m6A profiling may have their own superiorities and limitations, we compared the m6A sites detected by TGS (obtained by merging the m6A sites from ten ONT tools) to those detected by NGS (obtained by merging the m6A sites from miCLIP and miCLIP2). Even though the number of m6A sites from all ONT tools totaled 112,885, only 51.20% (19,239/37,577) of these sites that were detected by the NGS methods could be recalled (Fig. 7d). We further investigated the m6A sites detected by the NGS methods but were not recalled by any of the ONT tools and found that 15.51% were located in intergenic regions and 40.51% were covered by less than five ONT reads (Fig. 7d). After removing the intergenic and low-coverage sites, we re-analyzed the non-recall rates for sites with various sequence motifs, and found that m6A sites within the GGACT motifs exhibited a recall rate of more than 0.9 while AAACC and AAACA motifs exhibited recall rates of 0.3 (Fig. 7e). Since only a small proportion (19,239/112,885) of merged sites were supported by NGS methods, we investigated the characteristics of the remaining sites. These sites showed a clearer enrichment around the stop codons when supported by more ONT tools (Fig. 7f), while the overlap ratio of the m6A sites that were detected in mESCs KO samples decreased (Supplementary Fig. 13f). This implies that the NGS method failed to detect some of authentic m6A sites that were captured by the ONT tools.

## Discussion

With the development of NGS-based m6A detection methods, our understanding of the biological role of m6A has advanced considerably. However, such methods have inherent limitations. Recently, ONT DRS has emerged as a promising alternative method of detecting RNA modifications. To date, approximately ten computational tools have been developed. With the goal of comprehensively evaluating the performance of these tools, we conducted a comparative analysis of each tool's detection capabilities, thereby providing a guide for future studies using ONT DRS. Our analysis revealed a variable performance when evaluated using the ROC and PR curves as continuous metrics, with which m6Anet and Epinano_delta performed the best in the single-mode tools and compare-mode tools, respectively. Moreover, a similar ranking was observed when using different validation datasets or the union and intersection sets. We then applied these tools to the detected m6A sites under a cut-off which correlated to the highest F1 score and found that it was still challenging to balance precision and recall rates for some tools. For example, m6Anet, xPore and Nanocompore achieved the highest precision at the cost of lowest recall, while Tombo and Tombo_com achieved the opposite. Nevertheless, integrated predictions from all tools achieved better performance than individual tools, especially when using m6A sites which were supported by multiple tools (>4). As for quantitation, methylation rates reported by Tombo/Tombo_com, Nanom6A and xPore showed positive correlation to the enrichment score determined by MeRIP-seq, but were poorly agreed across ONT tools or with the methylation rates quantified by m6A-REF-seq. Taken together, there is still room for improvement in the algorithms for the identification and quantification of RNA modifications from ONT DRS datasets.

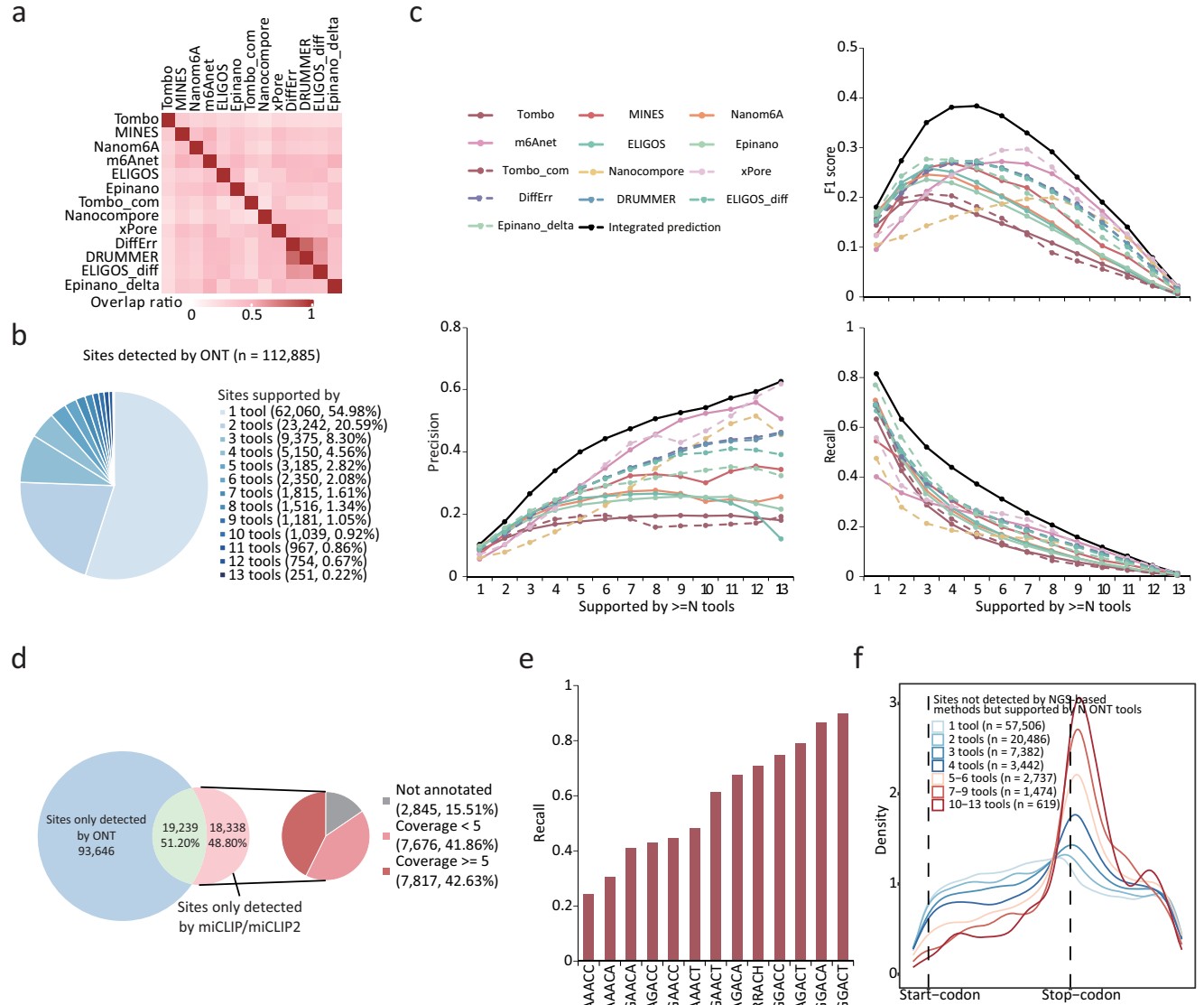

**Fig. 7 | Assessment of m6A detection by integrating results from all tools.**
**a** Recurrent sites ratio among top 5000 m6A sites detected by ONT tools. Heatmap shows the percentage of sites simultaneously detected by each pair of tools.
**b** Percentage of sites identified by multiple tools. **c** Precision, recall rate and F1 scores of sites supported by more than *N* tools versus identical number of top sites from individual tool. The validation set was derived from the miCLIP results.
**d** Merged sites detected across all tools and percentage of these sites found in the miCLIP/miCLIP2 results (left). Non-recalled sites were classified according to coverage (right). **e** Recall rates for m6A sites located in different motifs, using the merged sites detected across all tools after removing intergenic and low-coverage m6A sites. **f** Metagene plots of the transcriptome-wide distribution of m6A sites that could not be detected by NGS-based methods but supported by one or more ONT tools.

There are two main challenges that need to be addressed. The first challenge relates to the variance in electric current intensity observed during repeated sequencing of the same 5-mer motif, which is determined by the capability of ONT platform. Given that the presence of m6A corresponds to small differences in current intensity, large amounts of background variation could render A and m6A indistinguishable[37]. The second challenge relates to the data used to train machine learning models[38]. One common method to generate positive samples involves in vitro transcription with dm6ATP instead of dATP; however, the distribution of m6A on this artificial RNA may differ greatly from that found on a natural RNA. In addition, models trained on the DRS data containing organism-specific site information would have an inherent sequence bias. We believe synthetic positive samples with m6A sites located in predetermined locations with various motifs would be more appropriate[39].

We observed varying degrees of intrinsic bias towards some specific sites in various tools. Benefited from the use of nearly m6A-free negative control, compared-mode tools generally performed better than single-mode tools. Analogously, introducing a negative control (mESC *Mettl3* KO samples) as an extra calibrator effectively improved the precision of single-mode tools, although negative control sample with residual m6A confounded the analysis. However, it is relatively complicated to knockout *Mettl3* in most cell types[36,40,41] with some m6A sites having been shown to be *Mettl3*-independent[42], thereby making it virtually impossible to generate an ideal negative control from a natural sample. Recently, we used a transcriptome-wide modification-free RNA library synthesized through in vitro transcription (IVT) to systematically calibrate epitranscriptomic maps profiled by NGS based methods[16]. In the detection of RNA modifications via ONT DRS, this IVT RNA library holds great promise to serve as a gold standard negative control in the future.

Unexpectedly, all tools displayed very divergent recall rates and precision in different motifs. After excluding poorly sequenced reads and intronic or intergenic m6A sites, approximately 70% of the m6A sites detected by miCLIP/miCLIP2 were recalled from the merged m6A sites. High recall rates of up to 90% were seen in GGACT motifs, whereas lower rates of 20% were observed in AAACC motifs. Minor current differentials between modified and unmodified A's in some motifs may induce false-negative results. To overcome this bottleneck, chemical decorations would assist in making m6A more distinguishable from unmodified A when traversing the nanopore. Similar methods have been developed to detect inosine using the nanopore sequencing[43].

Here we represent the first comprehensive comparison of computational tools focused on detecting m6A in the ONT DRS dataset. Rational oligo models and well-established BS-seq data have already been applied to the benchmarking of 5mC detection by nanopore sequencing[44,45]. However, there is no gold-standard method for m6A detection with both single-base resolution and stoichiometric information on the NGS platform. Therefore, we conducted MeRIP-seq on the same samples used in the ONT sequencing and employed published m6A sets as identified by miCLIP, miCLIP2 and m6A-REF-seq. Though the value of evaluation metrics (ROC AUC, PR AUC, precision, recall, F1 score) given in this study may not reflect the performance of these tools completely and may be underestimated due to the low sequencing depth of DRS data, the comparison between tools would be informative. Moreover, most of our findings still hold when using different validation sets. In addition, we further observed that a proportion of m6A sites which were consistently captured by ONT tools were likely to be authentic, but were not detected by previous NGS-based methods. In conclusion, we provide a comprehensive comparison of computational tools commonly used in m6A mapping depending on the ONT DRS platform. Comparative information on the detection capability, intrinsic bias and motif preferences between the different tools provide valuable insights for the development of new detection strategies and algorithms in future.

## Methods

### Sample collection
The WT and *Mettl3* KO mESCs were validated in a previous study[46]. Briefly, exons 5–7 of *Mettl3* were replaced with puromycin and hygromycin resistance genes using CRISPR–Cas9 system. After selecting with 1 μg/ml puromycin (Gibco, A1113802) and 200 μg/ml hygromycin B (Sigma, V900372) for one week, diallelic KO colonies were cultured and verified by western blotting (see results in Liu et al.[46]). After extracting total RNA from the samples using TRIzol (Thermo Fisher Scientific, 15596026), mRNA was isolated via a Dynabeads mRNA Purification Kit (Thermo Fisher Scientific, 61006) with genomic DNA removal using TuRBO DNase kit (Thermo Fisher, AM2238).

### Nanopore direct RNA sequencing and data preprocessing
Nanopore direct RNA sequencing was conducted following instructions provided by Oxford Nanopore Technologies (Oxford, UK) using DRS kits (SQK-RNA002) and MinION flowcells (FLO-MIN106D, R9.4.1 pore). After live basecalling using Guppy (v4.0.11) in MinKNOW, reads that passed the quality threshold (7) were then subjected to post-run basecalling with the latest version of Guppy (v5.0.7) under default parameters. A reference transcriptome file was generated from the mouse genome reference and corresponding annotation (GENCODE.VM18) using convert2bed (v2.4.38) and bedtools (v2.26.0). The resulting FASTQ files were aligned to the mouse reference transcriptome using Minimap2 (v2.17) with parameter settings "-ax map-ont --MD". Samtools (v1.6) was used to filter secondary and supplementary alignments and convert aligned reads to the bam format. Public datasets were preprocessed in the same way except that *Arabidopsis* samples were aligned to the reference transcriptome generated from TAIR10 reference genome with Araport11 reference annotation.

### NGS-based validation sets
The raw MeRIP-seq data was downloaded from Zhang et al.[16] (GSE151028). After quality control, clean reads were aligned to the mouse genome reference file (GRCm38/mm10) with HISAT2 (v2.1.0) for peak calling using MACS2 (v2.1.2) (callpeak --keep-dup all -g mm --nomodel --extsize 200 -q 0.0001 --fe-cut-off 2). High confidence peaks were obtained by first merging peaks from two WT replicates and then subtracting peaks identified in both KO replicates. Only peaks containing RRACH motif were retained since most ONT tools were limited to this motif. Sites detected by ONT tools would be considered as true positives when they were found to be located in the peaks from MeRIP-seq. When using public miCLIP[34], miCLIP2[35] and m6A-REF-seq[12] datasets, m6A sites only in the RRACH motifs were kept. m6A profiles derived from miCLIP and miCLIP2 results were two main validation sets for all analysis. MeRIP-seq results were also used to compare tools with continuous evaluation metrics. When using MeRIP-seq data as evaluation set, we roughly considered all sites falling in the peaks as true positives. m6A-REF-seq results provided reference modification rate in the quantification part. As for the validation sets for *Arabidopsis*, we also filtered sites not containing RRACH motifs from the public MeRIP-seq[47] and miCLIP[28] datasets.

### m6A detection with multiple tools
We ran the tools with parameter settings described as follows, and the detailed command and code are available at https://github.com/zhongzhd/ont_m6a_detection[48].

**Tombo** (v1.5.1) needs to re-squiggle reads to assign raw current signals to each base before further processing can be carried out. Tombo now provides two methods for m6A detection: (1) a de novo non-canonical base method with the command "detect_modifications de_novo" (represented by Tombo); and (2) a canonical sample comparison method with the command "detect_modifications model_sample_compare" (represented by Tombo_com). The first approach was used for routine analysis of single samples, while the second approach was used to combine analyses for negative control samples. Several statistics were obtained from Tombo, including the coverage and estimated fraction-modified values, and the latter were treated as scores for ranking.

**MINES** uses coverage and fraction-modified values calculated by Tombo as input to its random forest models (only for AGACT and GGACH motifs) and only reports significant sites. Therefore, we modified the code to output the modification probability of all sites in all RRACH motifs. The modification probabilities of sites were treated as scores for ranking. The modified version is available at https://github.com/zhongzhd/ont_m6a_detection[48].

**Nanom6A** (v2.0) extracts the median, mean, standard deviation and dwell time from normalized raw signals for each read after re-squiggling with Tombo. The above features are then used as input for its XGBoost model which assigns a modification probability for each read. Reads with a modification probability >0.5 were considered modified, and the modified fraction of sites were treated as scores for ranking.

**m6Anet** (v1.0) requires the results generated from running the eventalign module in Nanopolish (0.13.2), which assigns raw current signal to each base (like Tombo). The results are used as input to the m6Anet's neural-network based Multiple Instance Learning model which produces probability-modified values per site. The probability-modified values corresponding to each site were treated as scores for ranking.

**Nanocompore** (v1.0.0) collapses the results from the Nanopolish eventalign module to generate a file containing the median intensity and dwell time; it also makes pairwise comparisons to the collapsed results from a control sample. The Gaussian mixture model (GMM) logit *p* values from statistical test results of sites were treated as scores for ranking. Note that this tool outputs many sites with "nan" value due

to failure in clustering, so sites with the "nan" values were assigned a $p$ value of 1.

**xPore** (v2.0) also uses the results from the Nanopolish eventalign module for all samples to make a configuration file, and applies the multi-sample two-Gaussian mixture model to obtain the estimated modification rate for each sample, a test statistic ($z$ score) and $p$ value on differential modification rates for each pairwise condition. The $z$ scores from statistical test results were treated as scores for ranking (as recommended by the authors). Similar to Nanocompore, this tool omitted many sites in its result due to failure in clustering, so we assigned a $z$ score of 0 to these sites. Sites with modified distributions of current assigned in the opposite direction, were assigned a $z$ score of 0 according to xPore's methods. In addition, sites with higher methylation rate in KO than WT were also assigned a $z$ score of 0.

**DiffErr** (v0.2) uses alignment files with a control as input. The $-\log_{10} p$ values from statistical test results of sites were treated as scores for ranking, and sites filtered internally or with odds ratio (log) $\leq 0$ were assigned a $-\log_{10} p$ value of 0.

**DRUMMER** also uses alignment files for the sample and a control as input. The $p$ values (odds ratio) from statistical test results of sites were treated as scores for ranking, and sites filtered internally or with odds ratio $\leq 1$ were assigned a $p$ value of 1.

**ELIGOS** (v2.0.1) offers two methods for the detection of modified nucleotides: (1) the identification of RNA modifications compared to a rBEM5 + 2 model using "rna_mod" (represented by ELIGOS); and (2) the identification of RNA modifications compared to control conditions using "pair_diff_mod" (represented by ELIGOS_diff). The first method was used for routine analyses of single samples, while the second was implemented for combined analyses including negative control samples. The $p$ values (odds ratio) from statistical test results of sites were treated as scores for ranking, and sites filtered internally or with odds ratio $\leq 1$ were assigned a $p$ value of 1.

**Epinano** (v1.2.0) offers two kinds of Support Vector Machine models for m6A detection. These models have different features: (1) the mean quality, and mismatch, insertion and deletion frequency of each base (represented by Epinano); and (2) the difference in mean quality, and in the mismatch, insertion and deletion frequency between the sample and a control for each base (represented by Epinano_delta). The first method was used by default and the second when comparing to a negative control. The probability-modified from predicted results were treated as scores for ranking.

### Tools comparison using ROC and PR curves

Sites in RRACH motif with more than 5 reads (except for m6Anet that required at least 20 reads) were tested by each tool and ranked according to the significant score as described above. We then converted the transcriptome coordinates to the genomic coordinates to match with the validation sets. Sites derived from multiple transcriptomic coordinates but aligned to the same genomic coordinate were merged as follows: for Tombo/Tombo_com, MINES, Nanom6A, m6Anet and Epinano/Epinano_delta, the scores (fraction-modified/probability-modified) were weighted averaging according coverage; for Nanocompore, xPore, DiffErr, DRUMMER and ELIGOS, the best scores ($p$ value/$z$ score) were selected as representative. To objectively compare the tools, we only kept the sites that were covered by all methods and with a coverage $\geq 20$ in the final dataset for evaluation using the ROC and PR curve. MeRIP-seq, miCLIP and miCLIP2 results mentioned above were applied as ground truth, and we also used the validation sets in the form of union and intersection of all the NGS-based methods. As for *Arabidopsis*, MeRIPseq and miCLIP results and their union and intersection were used as the validation sets.

### m6A determination with optimal cut-off

To determine the optimal cut-off for m6A detection, we varied the cut-offs and calculated F1 scores of sites in the RRACH motif covered by at least 5 reads using the MeRIP-seq, miCLIP and miCLIP2 results as validation sets separately (note that m6Anet requires at least 20 reads and MINES only output results in 4 motifs).

$$\text{F1 score} = \frac{2 \times \text{precision} \times \text{recall}}{\text{precision} + \text{recall}}$$

As the distribution of F1 scores were inconsistent between the validation set from MeRIP-seq and miCLIP/miCLIP2, we selected the cut-off corresponded to the maximum F1 score when adding up that of miCLIP and miCLIP2 for following m6A detection process. Validation sets of miCLIP and miCLIP2 were used for the following precision and recall calculation. Note that in evaluating the performance of these m6A detection tools, we only considered sites detected by NGS methods (MeRIP-seq, miCLIP and miCLIP2) that were covered by minimal DRS reads (5, 20, 50 DRS reads requirement). When setting the requirement at 5 DRS reads, we also took into consideration sites omitted by m6Anet (<20 reads) and MINES (not in four specific motif) and treated them with the lowest score for recall calculation. m6A sites detected under 5 DRS reads requirement were used for further analysis.

### Intrinsic bias assessment

Sites in the RRACH motif with minimum 5 reads (the pool) in *Mettl3* KO samples were tested by each tool and determined as m6A using the optimal cut-off described above. We then randomly sampled an identical number of sites (== the number of m6A sites detected in KO samples) from the pool as the random dataset for each tool. To test whether these tools prefer some irrelevant m6A sites (intrinsic bias), we compared two intersection "WT&KO" (m6A sites detected in WT and KO samples) and "WT&Random" (m6A sites detected in WT samples and existed in random dataset). If there were certain intrinsic bias, the WT&KO intersection should be greater than WT&Random intersection; we applied one-side Hypergeometric tests to calculate the significance. To determine for the cause of the intrinsic bias, the input data of MeRIP-seq were used to call SNPs/indels compared to mm10 reference using bcftools with the key parameter "QUAL > 100", and SNPs/indels were kept when supported by more than 2 of 4 samples (WT and KO with replicates). In addition, the flanking sequence (upstream and downstream 10 bp) of all candidate sites were extracted and inspected for homopolymer. We then compared the ratio of sites near SNPs/indels (within 5 bp) or homopolymer between the "WT&KO" intersection and sites in the RRACH motif with more than 5 reads (background) and applied one-sided Binomial tests to calculate significance.

### Comparisons of current intensities and base-calling "errors" between modified and unmodified 5-mer

Nanopolish eventalign was used to assign raw current to each base in the reference genome. The results of sam2tsv pileup (v0558422) were used to calculate the frequency of mismatch, deletion and insertion and base quality. Highly-methylated m6A sites in GGACA and GAACA motifs with a methylation rate differences >0.8 between WT and KO samples as determined by m6A-REF-seq, were selected. The current intensities and base-calling "errors" of these sites extracted from WT samples were treated as current intensities and base-calling "errors" of modified 5-mer, while these extracted from KO samples were treated as current intensities and base-calling "errors" of unmodified 5-mer. We compared the current intensities and base-calling "errors" between modified and unmodified 5-mer for both GGACA and GAACA motifs.

### m6A quantification

Tombo/Tombo_com and Nanom6A detect m6A at read level and support methylation rates calculation. Sites in the RRACH motif with more than 5 reads were calculated. xPore outputs methylation rates directly in its result and tests the difference of methylation rates

between WT and KO samples, so we used the methylation rates of WT samples for all sites in the RRACH motif with more than 5 reads except for sites which failed to cluster. Methylation rates from the m6A-REF-seq results were used for comparison. We also calculated the enrichment score for these sites using the MeRIP-seq data (IP FPKM/Input FPKM) with featureCounts (v2.0.1) for comparison.

## Reporting summary

Further information on research design is available in the Nature Portfolio Reporting Summary linked to this article.

## Data availability

All data generated for this paper have been deposited in the NCBI Gene Expression Omnibus (GEO) database under accession number GSE195618. Published DRS datasets derived from WT and *Mettl3* KO mESC samples are obtained from the work of Jenjaroenpun et al.[30] and are available through NCBI Sequence Read Archive (SRA) database under accession number SRP166020. Published Arabidopsis DRS datasets along with their KO variants are form Parker et al.[28] and are available through European Nucleotide Archive (ENA) database under accession number PRJEB32782.

## Code availability

The analysis pipeline and all custom scripts used for this paper are available at https://github.com/zhongzhd/ont_m6a_detection[48].

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

## Acknowledgements

This work was supported by the Ministry of Science and Technology of China (National Science and Technology Major Project, 2018YFA0109100, 2019YFA0802203, 2022YFA0912900, and 2022YFC3400400), National Natural Science Foundation of China (92253202, 32271499, 32270644, and 32100461), and Shenzhen Bay Scholars Program.

## Author contributions

G.-Z.L. conceived the project; Z.-D.Z. and Y.-Y.X. analyzed the data and wrote the manuscript; H.-X.C. conducted the experiments with the assistance from X.-H.L., J.-Y.J., Y.-L.L., F.W., L.J., and J.C.; Z.Z., D.W.M. and G.-Z.L. revised the manuscript. All authors reviewed the results and approved the final version of the manuscript.

## Competing interests

The authors declare no competing interests.
