## [Peer Review File · Nature Communications]

Systematic comparison of tools used for m6A mapping from nanopore direct RNA sequencingREVIEWER COMMENTS

Reviewer #1 (Remarks to the Author):

In the manuscript "Comprehensive benchmarking of tools used for m6A detection from nanopore direct RNA sequencing" the authors compare 10 different methods that detect m6A from Nanopore direct RNA-Seq data. The topic is highly relevant with many novel tools being developed that are evaluated with different data sets. An independent comparison of such tools has the potential to add value by highlighting strengths and weaknesses of the different approaches. The manuscript includes the most recent tools and presents a comprehensive overview that covers areas like prediction accuracy, recall, quantification, and potential sequence biases. Benchmarking of the methods is then done on a data set of wild type and METTL3 knockout mESCs, using MeRIP-Seq, m6A-REF-Seq, and miCLIP as ground truth. The work is well presented and very timely. There are some major limitations in the authors' approach that limit the validity of these results. However, if addressed, this manuscript would add valuable insights into a fast growing field.

Major comments:

1) Replace point estimates with continuous evaluation metrics

- The comparison of the different methods is based on single point estimates for a single threshold. While the authors use very stringent thresholds for some methods (eg Drummer), other methods seem to use very sensitive threshold settings (eg Tombo), which makes the results non-comparable. In addition, not all methods might describe default thresholds, in which case the results reflect the author's choice, but not the performance of the methods. To address this, the authors should evaluate methods across the full range of predictions using ROC curves and Precision-Recall curves.
- The current results based on single point estimates using different thresholds are helpful to demonstrate how default settings impact results, but they should not be interpreted to rank methods as more accurate or more sensitive. Currently, the results compare the default parameter settings, not the methods.
- if the authors would like to compare methods based on a point estimate (in addition to my suggestion above), I would suggest to select the same number of predicted sites based on a ranking (eg the top 1,000 sites)
- some methods might not return a score for ranking, in which case the authors could either run them with multiple different thresholds, or just add the single point estimate to the rank based comparison.
- The enrichment of m6A predictions at the 3'UTR is not useful as a comparison as this will favor methods with a stringent threshold, it reflects the thresholds more than the methods. In order to make this plot comparable, it should be generated based on an identical number of m6A predictions for each method (using the highest ranking predictions).

2) Include additional data

- The authors use direct RNA-Seq from a mouse cell line using wild type and METTL3 (loss of m6A) cells. The addition of this independent data set is important, however, the data set by itself is limited: the

dataset only consists of single replicates, and the data is sequenced with low throughput (MinION). Furthermore, the m6A data might not be of single nucleotide resolution. Additional replicates and high throughput sequencing (Promethion) will address many of the limitations that the authors highlight in this manuscript, and which might actually be limitations of the data not of the methods which are evaluated. To address this, the authors should use additional data sets such as the following which are already publicly available, and which contain additional m6a label data sets.

a) Synthetic data (<https://www.nature.com/articles/s41467-019-11713-9>)

b) METTL3 knockout data with multiple replicates for each condition (<https://www.nature.com/articles/s41467-019-11713-9>)

c) it might be fair to include data from another species as some methods have been developed and applied on non mammalian transcriptomes (e.g. <https://elifesciences.org/articles/49658>)

For the evaluation of quantification, the author rely on a procedure that has not been evaluated (the analysis of peaks). There are other data sets which provide ground truth data and which could be used here:

a) Synthetic data (<https://www.nature.com/articles/s41467-019-11713-9>)

c) Mixture experiments for quantification (<https://www.nature.com/articles/s41467-019-11713-9>)

3) Methods that require a control and which identify m6A without a control should be compared separately

- The accuracy for comparison based methods such as Nanocompore, Xpore, etc are expected to be higher since they utilize additional control samples that do not contain a lot of m6A. On the other hand, while single sample methods (Tombo, MINES, nanom6A, m6anet) might have fewer NGS methods support, they also have the advantage of not requiring additional KO samples which might not be easy or possible to generate. The application of methods that require a control and which make predictions without a control is different, and these methods should therefore be evaluated within each group.

- While a comparison of comparative and single sample methods is still helpful, this should not be interpreted in relation to the method, but in relation to the approach. Methods should be evaluated in comparison to methods using the same approach.

3) Ensure thresholds are used such that results are comparable

- the quantification part seems to evaluate methods across m6A levels from 0 to 100%. More stringent threshold will filter positions with low modification levels which will impact the quantification evaluation.

- the number of reads which are required to identify m6a sites might differ between the different methods, and not all sites in the m6A control data might be detectable. The authors could compare methods on a set of sites that are covered by all methods (eg minimum 25 reads)

- some methods like nanom6a or MINES are limited to only very few motifs, which will also be reflected in the recall/accuracy

4) Limitations in NGS based ground truth

- The authors claim that the trend in the model performance remains for all three NGS methods

independently, but it might be the case that some of these models are capturing authentic sites that do not really overlap between different NGS methods. It will be interesting to see if the “accuracy” or recall changes when we consider the validation set in the form of either union or intersection of all the NGS methods

- Maybe it is also worth looking at sites that are jointly captured by the DRS methods but not supported by the NGS technologies (since this will be considered false positives by the authors definition of accuracy and recall). For example, the DRS methods might capture sites that are hard to detect using the NGS technologies but this might not be false positives (this could be evaluated for example with the 3'UTR enrichment, or by comparison against knockout data)

5) The methods section could be improved with additional details, below are some examples:

- Currently, not all parameters or thresholds are listed.
- The analysis of knockout and random site predictions is not sufficiently clear.
- It's not clear how the NGS based m6a sites are compared to direct RNA-Seq sites as the NGS sites are not single base resolution. Are all sites that overlap considered true positives?
- How did the authors decide on the m6A sites used from NGS methods? Was a threshold applied?

6) The integrated analysis could be compared to the existing methods

- The integrated analysis returns results with higher accuracy when more methods overlap, which is expected. However, it's not clear if these sites are more accurate or less accurate than sites predicted by individual tools. In order to compare the integrated predictions to individual tool predictions, the authors could estimate precision and recall for each set (sites from 10 tools, sites from 9 tools etc) and compare that to the precision and recall obtained from each individual method for the same number of sites.
- Some methods only make predictions on a subset of possible m6A motifs (for example, nanom6a detects 12 motifs while MINES only detects 4 motifs). It's expected that the most nanom6a can overlap with MINES is only among the 4 motifs, so the top predictions with support from all methods are also limited to these motifs, which is a major limitation.

7) some claims are not supported by data or unclear without context, for example:

- "balancing accuracy and recall is challenging for tools": as the authors only compared single thresholds, the balance between accuracy and recall was not explored. The authors seem to imply that other (non direct RNA-Seq) methods might achieve better balance between accuracy and recall, however there is no data to show that, so this claim is highly suggestive and quantified
- "quantification was not reliable": it's not clear how reliable the ground truth data is that was used here, so this claim also is not supported by data.
- the meaning of "intrinsic bias" is not very clear
- "In conclusion, error-based methods had higher accuracy than current-based methods, but suffered from much lower recall rates." This reflects thresholds used, not the methods.
- "huge room for improvement": this is not clear from the data, it could also be a limitation of the NGS

based methods which might have missed valid sites

Minor:

- Accuracy does not seem to be the right term. Here: Accuracy = ratio of authentic m6A sites to all candidate sites (authentic if identified by both ONT tool and NGS based methods); Recall = ratio of peaks / sites containing authentic m6A sites / all peaks or sites in the validation dataset. The term accuracy here feels misleading, precision is a better term since this only measures the percentage of detected sites by each tool that is supported by the NGS based methods (i.e. how many % of the predicted positives are actually positives) (see for example: https://en.wikipedia.org/wiki/Precision_and_recall)

- "Sites detected in KO samples did not show enrichment around stop codons" but the supplementary Fig. 3a-b seems to indicate some enrichment near the stop codon?

- In the section that analyses predictions in the knockout sample, it's unclear why comparative methods should benefit when sites are removed that are detected in the knockout. I would expect that each site can only be detected in either the knockout or in the wild type, but not in both at the same time. I understand for the methods that do not use the knockout why that works, but for the comparative methods this is not clear, and should theoretically not be possible

-the recall analysis for each kmer: even with miCLIP/NGS data, the FP rate might differ by kmer, with GGACT having less false positives than GAACA for example, does that influence these results?

- This sentence is not clear: "After considering the resolution and limitations of different NGS-based methods, we used MeRIP-seq, miCLIP and miCLIP2 results as validation sets."

-Figure 3 a is not clear, what are the different plots? What is the meaning of random here? this is again a comparison of thresholds

-Figure 5b: axes labels could be added, it's difficult to understand currently what is shown

Reviewer #2 (Remarks to the Author):

In this manuscript, Zhong et al did a comprehensive comparison of ten published tools for the m6A detection from the nanopore direct RNA sequencing data. They evaluated the performance (i.e. accuracy, recall, and quantification) of each tool with data from the mRNA of Mettl3 WT/KO mESC cells and assessed the intrinsic bias. Overall, the work identified many weaknesses of the established methods and provided several insights for better performance. However, the authors would need more

efforts to have evident contributions to the biological insights and provide clarifications for some major issues below:

1. In Fig. 1, authors generated the Mettl3 KO mESC as the negative control. The loss of m6A was confirmed by LS-MS/MS (Fig. 1b) and MeRIP-seq (Fig. S1c). The authors would need to disclose the strategy for the Mettl3 KO. Furthermore, more details would be needed to show the complete loss of METTL3 protein or its loss of function in the cells. As recently reported, most published METTL3 mutant cell lines have not achieved complete loss function of METTL3 (<https://www.biorxiv.org/content/10.1101/2021.12.15.472866v1>).
2. To evaluate the soundness of the authors' data analysis, the readers would appreciate if the authors could share the raw code for the m6A detection with each tool. If it's possible, a platform (e.g. as a docker or singularity image) that could run all of the ten tools for m6A detection would greatly help the scientific community. Moreover, the authors should disclose the code of the data analysis for readers (e.g. at github).
3. Since MeRIP-seq didn't have single nucleotide resolution, how the sites obtained by nanopore could be validated by MeRIP-seq in Fig.2?
4. The authors need to provide adequate details for the rationale of the candidate pool of RRACH sites (line228), and how these sites are selected as the candidate pool? Even though calibrating m6A sites with KO samples could improve the accuracy, the difference is not significant for most of the tools. Could the authors show the difference between the true-positive and false-positive sites? For example, the motifs around the sites, nucleotide ratio. It might help to explain the causes of the bias.
5. It's known that GGACT, GGACA, GGACC, and AGACT are more likely to be methylated. The minor current differentials between methylated and unmethylated As in other motifs may be the result of their low methylation levels by nature, while not the sensitivity of the current differentials. A direct RNA nanopore sequencing of the in-vitro synthesized methylated or unmethylated oligos containing these RNA motifs would provide a clean experiment to test this hypothesis.
6. As m6A-REF-seq only quantified a subset of m6A sites (RRACA sites), miCLIP sites should be used as it covers much more m6A sites in single nucleotide resolution and the peak enrichment of miCLIP sites reflects the methylation levels of the miCLIP sites.

Reviewer #3 (Remarks to the Author):

Zhong et. al. presents a study on comparison m6A identification of different bioinformatic tools from direct RNA sequencing of Oxford Nanopore Technology with the m6A sites identified by antibody-NGS based technique of MeRIP-seq. The comparisons were further extended to compare the m6A sites with m6A-REF-seq, miCLIP and miCLIP2 method from independent studies. The comparisons were conducted with the transcriptome of the well-known biological case of WT vs Mettl3-knockout to study m5A in mESCs.

The study has a good objective to evaluate the performance of the bioinformatic tools to identify m6A from native RNA sequences at signal levels or errors levels. However, there are many points that need to

be addressed in the manuscript to make fair comparison.

Major Comments.

- Benchmarking needs the ground truth to reference. The authors used the high confident m6A sites identified from MeRIP-seq to compare. I don't think, this is the ground truth even though the authors published their improvement on calling m6A using NGS in recent published paper(<https://www.nature.com/articles/s41592-021-01280-7.pdf>). So "benchmarking" should not be used, because we all know that there is certain level of false positive remains in the identified m6A sites. I strongly recommend the authors use "comparison" instead of "benchmarking".

- Among NGS-base methods, the m6A sites under constraints of RRACH motif identified from MeRIP-seq is consistent with m6A-REF-seq or miCLIP or not? This is a very important information that need to firstly clarify. The author showed some indirect comparison at the end (sup. Fig. 7). It is important to show first comparison result among NGS-based method and present a rationale selection of m6A sites under constraints of RRACH motif as the set to compare with DRS-based methods. From the sup. fig7, The number of identified m6A sites are approximately 18,000, 13,000 and 2,000 for MeRIP-seq, or miCLIP and m6A-REF-seq, respectively. I can see that there are several disagreements among NGS-based methods! How we can ensure the results from MeRIP-seq, which are peaks not single nucleotides is gold standard??. Should the author use combination with evident ranking (how many time the identified m6A sites present in the three methods) or intersection of all m6A sites derived from MeRIP-seq, or miCLIP and m6A-REF-seq ?? This comment is very important point that need to be clearly presented at first. The small number of m6A sites identified by m6A-REF-seq resulted from low sequencing depth??

- Sequencing depth is always the important factors in calling m6A sites in either NGS-based or DRS-based methods. With the various level of expression of different transcripts, I doubt that ~1.3 million reads of generated from DRS would be sufficient to make comparison especially in low expressed transcripts? The author also reported the small fraction of reads are full length in sup.fig. 1b due to the strong 3' bias of DRS. To have a fair comparison between sequencing methods, the author should have the similar amount of data for NGS and DRS to begin with. We know that the NGS-Based methods have much better resolution because we amplified and sequenced only the DNA fragments derived from enriched region not whole transcriptome as DRS-based methods. Therefore, we need to work on this issue before performing the comparison between NGS-Based and DRS-based methods. It can be done for example by simple calculations of the sequencing depth of at all RRACH motifs then use the information to normalize the sequencing depth. The mean transcript coverage depth is not a good indicator, it is a corpse estimate. This issue recently reported in the studies of Grünberger et.al. (<https://rnajournal.cshlp.org/content/early/2021/12/14/rna.078937.121>) and and Wongsurawat et.al. (<https://www.frontiersin.org/articles/10.3389/fbioe.2022.842299/full>) in bacteria and yeast system. The 3'bias of DRS is a major factor that impact the RNA modification callings therefore the authors need to make sure about sequencing depth at individual RRACH motif.

- After clear the sequencing depth issues above, cut off is the next step. Default cut-off is an intuitively recommendation by the developer that may not be the optimum choice as the research field of identification of RNA modification from native sequences/signals is still not mature yet. As the authors have a good dataset, I recommend that authors performed sensitivity analysis of the cut off and recommend to the readers. This will be a very important results for the field. For example, the authors

vary cut-offs and calculate the performance matrix such as F1score (harmonic mean of precision and recall) or accuracy and then choose the best cutoff for further comparisons. These will be very useful information to the broad audiences.

- The stoichiometry of m6A is another major factor in calling methylation from DRS. The tools that need reads population as input will suffer on low stoichiometry of m6A sites. On the other hand, read level identification could handle the low stoichiometry m6A sites. Therefore, the comparison of the performance in bins of stoichiometry levels as shown in Fig. 7 is a good approach. However, the number of m6A sites identified by m6A-REF-seq is small. The result may not reflect the overall picture. This is a limitation of this result.

- Integrated analyses section as Fig. 7 shows the comparison of DRS vs NGS by combining the m6A sites identified by 10 tools. As the earlier results show the low overlap between NGS-based and DRS-based method, I don't think this comparison will gain information for the reader. Authors should not do this.

- For reproducibility of the result, which is a serious issue in computational biology research, the authors NEED to provide a computational notebook containing codes, commands and descriptions of how all bioinformatic analyses were performed for the readers.

Minor comments

Line 66 "PacBio sequencing" -> "PacBio DNA sequencing?". To make sure that the detection of DNA and RNA modification is accomplished by DNA sequencing.

Nanopolish needs to be in the timeline of the Sup. Fig. 1a. The timeline of the tool should be based on the publication date of the major paper of the tool.

Point-by-point response to reviewer comments

Summary

We are very grateful for the constructive comments and suggestions from three reviewers. We revised the manuscript and updated figures accordingly. We also refined the language with the help of a native speaker. We believe that all the concerns and questions raised by the reviewers have now been well-addressed. Please check the following point-by-point response. Our replies are **highlighted**, and the modified text/figures in the manuscript are also **marked**.

Reviewer #1 (Remarks to the Author):

In the manuscript "Comprehensive benchmarking of tools used for m6A detection from nanopore direct RNA sequencing" the authors compare 10 different methods that detect m6A from Nanopore direct RNA-Seq data. The topic is highly relevant with many novel tools being developed that are evaluated with different data sets. An independent comparison of such tools has the potential to add value by highlighting strengths and weaknesses of the different approaches. The manuscript includes the most recent tools and presents a comprehensive overview that covers areas like prediction accuracy, recall, quantification, and potential sequence biases. Benchmarking of the methods is then done on a data set of wild type and METTL3 knockout mESCs, using MeRIP-Seq, m6A-REF-Seq, and miCLIP as ground truth. The work is well presented and very timely. There are some major limitations in the authors' approach that limit the validity of these results. However, if addressed, this manuscript would add valuable insights into a fast growing field.

Response: We appreciate this reviewer's positive comments. According to these constructive suggestions, we updated the analysis to enhance the validity and rigidity of our results. Now we believe this manuscript has been substantially improved.

Major comments:

1) Replace point estimates with continuous evaluation metrics

- The comparison of the different methods is based on single point estimates for a single threshold. While the authors use very stringent thresholds for some methods (eg Drummer), other methods seem to use very sensitive threshold settings (eg Tombo), which makes the results non-comparable. In addition, not all methods might describe default thresholds, in which case the results reflect the author's choice, but not the performance of the methods. To address this, the authors should evaluate methods across the full range of predictions using ROC curves and Precision-Recall curves. The current results based on single point estimates using different thresholds are helpful to demonstrate how default settings impact results, but they should not be interpreted to rank methods as more accurate or more sensitive. Currently, the results compare the default parameter settings, not the methods.

Response: We appreciate a lot for this suggestion. In our original analysis, we compared the performance of these tools by their default parameter settings because we thought users may adopt the default settings in most cases. However, as this reviewer pointed out,

these results reflected the developers' and our choice, but not the performance of the tools. And point estimates do not provide comprehensive information, especially when default parameters are not optimal for the current datasets. We thus replaced point estimates with continuous evaluation metrics including Receiver Operating Characteristic (ROC) and Precision Recall (PR) curves (Fig. 2a-d and Supplementary Fig. 2a-b). Now the results should be more comprehensive and convincing.

- if the authors would like to compare methods based on a point estimate (in addition to my suggestion above), I would suggest to select the same number of predicted sites based on a ranking (eg the top 1,000 sites)

Response: We thank this reviewer and think this suggestion is reasonable and helpful. In the updated version, we calculated the Precision of top 2,000, 4,000, ..., 20,000 sites predicted by different tools (Fig. 2e, g and Supplementary Fig. 2c, e). We think that this information will help users to choose the most appropriate tool with the highest precision when they have an estimation of the amount of m6A sites, or they expect to obtain a fixed number of m6A sites.

- some methods might not return a score for ranking, in which case the authors could either run them with multiple different thresholds, or just add the single point estimate to the rank based comparison.

Response: We thank this suggestion. Fortunately, all tools report a ranking score representing the information of modification fraction, modification probability, and degree of significance. We used this score in the further analysis.

- The enrichment of m6a predictions at the 3UTR is not useful as a comparison as this will favor methods with a stringent threshold, it reflects the thresholds more than the methods. In order to make this plot comparable, it should be generated based on an identical number of m6A predictions for each method (using the highest ranking predictions).

Response: We thank this reviewer for this reasonable suggestion. We have updated the metagene plots and compared the distribution of the top 2,000 m6A sites detected by each tool (Fig. 2i-j).

2) Include additional data

- The authors use direct RNA-Seq from a mouse cell line using wild type and METTL3 (loss of m6A) cells. The addition of this independent data set is important, however, the data set by itself is limited: the dataset only consists of single replicates, and the data is sequenced with low throughput (MinION). Furthermore, the m6A data might not be of single nucleotide resolution. Additional replicates and high throughput sequencing (Promethion) will address many of the limitations that the authors highlight in this manuscript, and which might actually be limitations of the data not of the methods which are evaluated. To address this, the authors should use additional data sets such as the following which are already publicly available, and which contain additional m6a label data sets. a) Synthetic data (<https://www.nature.com/articles/s41467-019-11713-9>) b) METTL3 knockout data with multiple replicates for each condition

(<https://www.nature.com/articles/s41467-019-11713-9>) c) it might be fair to include data from another species as some methods have been developed and applied on non-mammalian transcriptomes (e.g. <https://elifesciences.org/articles/49658>)

Response: We appreciate this constructive suggestion. Accordingly, we have included a replicate dataset derived from wild-type and *Mettl3* knock-out (KO) mouse embryonic stem cells (Jenjaroenpun, P. et al. <https://doi.org/10.1093/nar/gkaa620>). We found that all single-mode tools achieved nearly identical performance to that using our data; however, tools in the compare-mode generally performed worse while the performance ranking remained the same (Supplementary Fig. 4). These results meet our expectation because the published mESC was found to be an incomplete *Mettl3* KO cell line with an estimated ~40% of m6A still present (Batista, P. J. et al. <https://doi.org/10.1016/j.stem.2014.09.019>), while our KO sample has been proved to be virtually absent of m6A. It also implies that a perfect negative control absent of any m6A would be crucial for these compare-mode tools. For additional datasets this reviewer suggested:

a) Synthetic data (<https://www.nature.com/articles/s41467-019-11713-9>)

This synthetic dataset was used to train models for Nanom6A and EpiNano, so we think it could be unfair to apply the comparing analysis with other tools. Furthermore, this dataset was generated using *in vitro* transcription with all the As replaced with m6As. This kind of artificial sequences are quite distinct from native RNA and we think the dataset is not a good reference for comparison analysis.

b) METTL3 knockout data with multiple replicates for each condition (<https://www.nature.com/articles/s41467-019-11713-9>)

Three independent biological replicates from wild type and *IME4 (Mettl3)* KO yeast were collected in this article, however, the number of m6A sites is too small in yeast to evaluate tools.

c) it might be fair to include data from another species as some methods have been developed and applied on non-mammalian transcriptomes (e.g. <https://elifesciences.org/articles/49658>)

We agree that including a non-mammalian specie (such as *Arabidopsis*) would be helpful to evaluate tools more comprehensively. Previous NGS-based methods identified nearly 5,000 m6A in *Arabidopsis*. We applied the comparative analysis using the DRS datasets derived from *Arabidopsis* and found some features different from mammals: the most dominant motif in *Arabidopsis* is AAACH, rather than AGACT and GGACH in human or mouse (Supplementary Table 4). It also explains the observation that the performance of machine-learning based tools, particularly m6Anet and MINES which are trained on DRS data from human samples using miCLIP as the ground truth, are compromised when applying to the *Arabidopsis* dataset (Supplementary Fig. 5). We also found almost all error-based tools performed better in *Arabidopsis* (Supplementary Fig. 5), which would likely benefit from a higher coverage (Supplementary Table 3). Accordingly, given that the size of *Arabidopsis* genome is only 1/20 compared to that of the mouse genome, a substantially higher coverage was achieved in *Arabidopsis* sample under similar sequencing reads (Supplementary Table 3), which meant that more base-calling "errors" occurred in the presence of m6A and increased the statistical power of error-based tools. Notably, as more than 10% of m6A were retained in *vir-1* samples, the performance of the

compare-mode did not significantly improve (Supplementary Fig. 5). We observed exactly the same result from another replicate of Arabidopsis (Supplementary Fig. 7).

- For the evaluation of quantification, the author rely on a procedure that has not been evaluated (the analysis of peaks). There are other data sets which provide ground truth data and which could be used here:

a) Synthetic data (<https://www.nature.com/articles/s41467-019-11713-9>)

b) Mixture experiments for quantification

(<https://www.nature.com/articles/s41467-019-11713-9>)

Response: We agree with this comment and also think “ground truth” data would be essential. However as mentioned earlier, this kind of synthetic sequences are quite distinct from native RNA and we think the dataset is not a good reference for comparison analysis. Also, this synthetic dataset was used to train models for Nanom6A, so we think it could be unfair to apply the comparing analysis with other tools. Tombo/Tombo_com and Nanom6A detect m6A at individual read, thus their quantification ability could not be evaluated by mixtures of methylated and unmethylated reads as the previous study did (<https://www.nature.com/articles/s41467-019-11713-9>). As a compromise, we now used the quantification results from m6A-REF-seq as ground truth (Fig. 5b). Given that m6A sites identified by m6A-REF-seq are limited to ACA motifs, we also calculated the sites enrichment scores ($FPKM_{IP} / FPKM_{input}$) from MeRIP-seq data to represent the m6A rate (Fig. 5a).

3) Methods that require a control and which identify m6A without a control should be compared separately

- The accuracy for comparison based methods such as Nanocompore, Xpore, etc are expected to be higher since they utilize additional control samples that do not contain a lot of m6A. On the other hand, while single sample methods (Tombo, MINES, nanom6A, m6anet) might have fewer NGS methods support, they also have the advantage of not requiring additional KO samples which might not be easy or possible to generate. The application of methods that require a control and which make predictions without a control is different, and these methods should therefore be evaluated within each group.

- While a comparison of comparative and single sample methods is still helpful, this should not be interpreted in relation to the method, but in relation to the approach. Methods should be evaluated in comparison to methods using the same approach.

Response: We thank this reviewer and realize that whether requiring a control is the key difference between these tools, and it also influences the user choice. In this revised version, we classified the tools according to this feature and compared them separately (Fig. 2 and Supplementary Fig. 2).

3) Ensure thresholds are used such that results are comparable

- the quantification part seems to evaluate methods across m6A levels from 0 to 100%. More stringent threshold will filter positions with low modification levels which will impact the quantification evaluation.

Response: We apologize for the confusion but we did not filter sites based on the

threshold in the quantification part. However, Tombo got more sites evaluated because of its lower coverage requirement (minimum 5 reads by default) than Nanom6A (minimum 20 reads by default) and Xpore (minimum 15 reads by default). Now we have made the quantification evaluation of all sites under the same coverage requirement to make this comparison fair (minimum 5 reads) (Fig. 5).

- the number of reads which are required to identify m6a sites might differ between the different methods, and not all sites in the m6A control data might be detectable. The authors could compare methods on a set of sites that are covered by all methods (eg minimum 25 reads)

Response: We appreciate this suggestion. In the revised version, we compared methods on a set of sites that are covered by all methods. In the part in which we evaluated tools using ROC AUC and PR AUC, we compared tools using the sites covered by all tools with minimum 20 reads. In the following part for m6A detection under cut-offs, we retained sites with minimum 5 reads.

- some methods like nanom6a or MINES are limited to only very few motifs, which will also be reflected in the recall/accuracy

Response: We thank this reviewer for pointing out. We also realized this issue. In the part in which we evaluated tools using ROC AUC and PR AUC, we limited our evaluation on sites in RRACH motif. One exception is MINES which trains model using RRACH motifs but originally outputs sites only in 4 motifs (GGACT, GGACA, GGACC, AGACT). We arbitrarily modified its code to permit the output in other motifs and stated in the manuscript. In the following part for m6A detection under cut-offs, all sites in RRACH motifs with minimum 5 reads were considered. For recall calculation, sites omitted by m6Anet (< 20 reads) or MINES (not in the four specific motifs) were also included and assigned with minimum score. It is natural to speculate that some tools gain a trade-off between precision and recall, and high coverage requirement or limited motifs would improve precision while lose recall.

4) Limitations in NGS based ground truth

- The authors claim that the trend in the model performance remains for all three NGS methods independently, but it might be the case that some of these models are capturing authentic sites that do not really overlap between different NGS methods. It will be interesting to see if the “accuracy” or recall changes when we consider the validation set in the form of either union or intersection of all the NGS methods

Response: We entirely agree that none of the current NGS-based method could be gold standard. Therefore, we adopted multiple independent validation sets to evaluate these tools separately. Using the validation sets in the form of either union or intersection of all the NGS methods is a good strategy to avoid the influence of false negatives and false positives. We have added this analysis in the part of performance comparison (Fig. 2b, d and Supplementary Fig. 2a-b).

- Maybe it is also worth looking at sites that are jointly captured by the DRS methods but

not supported by the NGS technologies (since this will be considered false positives by the authors definition of accuracy and recall). For example, the DRS methods might capture sites that are hard to detect using the NGS technologies but this might not be false positives (this could be evaluated for example with the 3'UTR enrichment, or by comparison against knockout data)

Response: We thank this reviewer for this constructive suggestion. It would be an interesting attempt to compare NGS and DRS in m6A detection, though we think the current data are not sufficient to draw any conclusion. In the revised version, we added the evaluation of these "false positives" sites, and we found that these sites showed clearer enrichment around the stop codon when supported by more ONT tools (Fig. 7f), while the overlap ratio of the m6A sites that were also detected in mESCs KO samples decreased (Supplementary Fig. 13f). This implies that the NGS methods fail to detect some of authentic m6A sites that are captured by the ONT tools.

5) The methods section could be improved with additional details, below are some examples:

- Currently, not all parameters or thresholds are listed.
- The analysis of knockout and random site predictions is not sufficiently clear.

Response: We thank this reviewer for pointing out. We have added more details in the revised Methods section.

- It's not clear how the NGS based m6a sites are compared to direct RNA-Seq sites as the NGS sites are not single base resolution. Are all sites that overlap considered true positives?

Response: The main validation datasets are generated by miCLIP and miCLIP2, which detect m6A sites at single-base resolution. When using MeRIP-seq data, we roughly considered the sites falling in the peaks as true positives. We think this process is reasonable because multiple m6A sites tend to cluster as MeRIP-seq peaks (Linder et al., *Nat Methods*. **12**, 767–772, <https://doi.org/10.1038/nmeth.3453>). Though it may overestimate the precision and recall of ONT methods, we speculate that the ranking of performance would not change (Fig. 2b, d and Supplementary Fig. 2a-b).

- How did the authors decide on the m6A sites used from NGS methods? Was a threshold applied?

Response: We adopted a canonical pipeline to call m6A peaks from MeRIP-seq data (see Methods), and then merged peaks from two WT replicates. After subtracting peaks that were also identified in KO replicates, we obtained high confidence peaks and reserved those containing RRACH motifs. As for miCLIP and miCLIP2, we confirmed the published analysis pipeline and downloaded the processed results, then we filtered sites not in RRACH motifs (see Methods).

6) The integrated analysis could be compared to the existing methods

- The integrated analysis returns results with higher accuracy when more methods overlap, which is expected. However, it's not clear if these sites are more accurate or less accurate

than sites predicted by individual tools. In order to compare the integrated predictions to individual tool predictions, the authors could estimate precision and recall for each set (sites from 10 tools, sites from 9 tools etc) and compare that to the precision and recall obtained from each individual method for the same number of sites.

Response: We appreciate this constructive suggestion from this reviewer. We compared the precision, recall and F1 scores of the integrated predictions to that of the same number of top m6A sites from each individual tool. We found that the integrated predictions achieved a better performance compared to individual tools. Moreover, using m6A sites supported by more than 4 or 5 tools achieved the highest F1 score (Fig. 7c and Supplementary Fig. 13a-e), thus highlighting the effectiveness of integrating the predictions from multiple tools.

- Some methods only make predictions on a subset of possible m6A motifs (for example, nanom6a detects 12 motifs while MINES only detects 4 motifs). It's expected that the most nanom6a can overlap with MINES is only among the 4 motifs, so the top predictions with support from all methods are also limited to these motifs, which is a major limitation.

Response: We thank this reviewer for pointing out this issue. We revised this part and found that the integrated predictions achieved a better performance compared to individual tools. Moreover, using m6A sites supported by more than 4 or 5 tools achieved by far the highest F1 score (Fig. 7c and Supplementary Fig. 13a-e).

7) some claims are not supported by data or unclear without context, for example:

- "balancing accuracy and recall is challenging for tools": as the authors only compared single thresholds, the balance between accuracy and recall was not explored. The authors seem to imply that other (non direct RNA-Seq) methods might achieve better balance between accuracy and recall, however there is no data to show that, so this claim is highly suggestive and quantified

Response: We thank this reviewer for pointing out this issue. We revised the manuscript to "most tools presented a trade-off between precision and recall". Importantly, we followed this reviewer's suggestion and performed comparison with continuous evaluation metrics (Fig. 2a-d, Fig. 2a-b and Supplementary Fig. 8).

- "quantification was not reliable": it's not clear how reliable the ground truth data is that was used here, so this claim also is not supported by data.

Response: We agree with this reviewer. Currently there is no gold standard for m6A quantification. We deleted this statement in the revised manuscript.

- the meaning of "intrinsic bias" is not very clear

Response: We found that some ONT tools detected substantial m6A sites in both *Mettl3* KO and WT samples (Fig. 3c). Therefore, we speculated some ONT tool may favor certain sites irrelevant to m6A, which could be interpreted as the "intrinsic bias" of these tools. Further analysis revealed that this kind of bias was partially due to SNPs/indels and homopolymer (Supplementary Fig. 9d-e), which could introduce alignment errors in Nanopolish-eventalign and Tombo-resquiggle. We revised this section to make it clearer.

- "In conclusion, error-based methods had higher accuracy than current-based methods, but suffered from much lower recall rates." This reflects thresholds used, not the methods.
Response: We agree with this reviewer and deleted this statement in the revised manuscript.

- "huge room for improvement": this is not clear from the data, it could also be a limitation of the NGS based methods which might have missed valid sites
Response: We thank this reviewer for pointing out. We deleted this statement and revised our manuscript thoroughly.

Minor:

- Accuracy does not seem to be the right term. Here: Accuracy = ratio of authentic m6a sites to all candidate sites (authentic if identified by both ONT tool and NGS based methods); Recall = ratio of peaks / sites containing authentic m6A sites / all peaks or sites in the validation dataset. The term accuracy here feels misleading, precision is a better term since this only measures the percentage of detected sites by each tool that is supported by the NGS based methods (i.e how many % of the predicted positives are actually positives) (see for example: https://en.wikipedia.org/wiki/Precision_and_recall)
Response: We thank this reviewer for pointing out. We realized the right term we intended to express should be "precision" rather than "accuracy". Now it has been corrected all through the manuscript.

- "Sites detected in KO samples did not show enrichment around stop codons" but the supplementary Fig. 3a-b seems to indicate some enrichment near the stop codon?
Response: We thank this reviewer for pointing out. We also noticed that sites detected in KO samples show slight enrichment around stop codons, presumably due to the 3' bias of DRS. To make it clear, we added the distribution plot of background control to the original figure (Supplementary Fig. 9c).

- In the section that analyses predictions in the knockout sample, it's unclear why comparative methods should benefit when sites are removed that are detected in the knockout. I would expect that each site can only be detected in either the knockout or in the wild type, but not in both at the same time. I understand for the methods that do not use the knockout why that works, but for the comparative methods this is not clear, and should theoretically not be possible
Response: We thank this reviewer for this insightful question. In this section, we would like to explore if there were "intrinsic bias" of ONT tools in detecting m6A, and we applied the tools to both the WT and KO samples. Indeed, this comparison could only be applicable for tools that do not require a control, including Tombo, MINES, Nanom6A, m6Anet, ELIGOS and EpiNano. We originally included Xpore, since this tool also supports outputting methylation rate for both WT and KO samples. To make it clear, we removed Xpore in the revised manuscript.

-the recall analysis for each kmer: even with miCLIP/NGS data, the FP rate might differ by kmer, with GGACT having less false positives than GAACA for example, does that influence these results?

Response: We thank this reviewer for pointing out. We also noticed this issue. Some tools may detect more sites in GGACH/AGACT motifs and the corresponding recall rates would be higher. In the revised version, we added the precision analysis for each kmer, and we found that the performances of most tools for the sites in GGACH/AGACT motifs were more sensitive and more precise (Fig. 4a-b, and Supplementary Fig. 10a-b).

- This sentence is not clear: "After considering the resolution and limitations of different NGS-based methods, we used MeRIP-seq, miCLIP and miCLIP2 results as validation sets."

Response: We thank this reviewer for pointing out. We realized that the original description was vague. We rewrote it as "In considering the limitations of each NGS-based methods, we took the m6A sites identified by each NGS-based method or their intersection and union as validation sets separately".

-Figure 3a is not clear, what are the different plots? What is the meaning of random here? this is again a comparison of thresholds

Response: We thank this reviewer for pointing out. We revised the figure (Figure 3c in the current version) and added more details in main text and Methods section: Sites in the RRACH motif with minimum 5 reads (the pool) in *Mettl3* KO samples were tested by each tool and determined as m6A using the optimal cut-off described above. We then randomly sampled an identical number of sites from the pool as the random dataset for each tool. To test whether these tools prefer some m6A irrelevant sites (intrinsic bias), we compared two intersection "WT&KO" (m6A sites detected in both WT and KO samples) and "WT&Random" (m6A sites detected in WT samples and also in random dataset).

-Figure 5b: axes labels could be added, it's difficult to understand currently what is shown

Response: We thank this reviewer for pointing out. We added axes labels in the revised version.

Reviewer #2 (Remarks to the Author):

In this manuscript, Zhong et al did a comprehensive comparison of ten published tools for the m6A detection from the nanopore direct RNA sequencing data. They evaluated the performance (i.e. accuracy, recall, and quantification) of each tool with data from the mRNA of *Mettl3* WT/KO mESC cells and assessed the intrinsic bias. Overall, the work identified many weaknesses of the established methods and provided several insights for better performance. However, the authors would need more efforts to have evident contributions to the biological insights and provide clarifications for some major issues below:

Response: We appreciate this reviewer's positive comments. According to the constructive suggestions, we updated the analysis to enhance the validity and rigidity of our results. Now we believe this manuscript has been substantially improved.

1. In Fig. 1, authors generated the Mettl3 KO mESC as the negative control. The loss of m6A was confirmed by LS-MS/MS (Fig. 1b) and MeRIP-seq (Fig. S1c). The authors would need to disclose the strategy for the Mettl3 KO. Furthermore, more details would be needed to show the complete loss of METTL3 protein or its loss of function in the cells. As recently reported, most published METTL3 mutant cell lines have not achieved complete loss function of METTL3 (<https://www.biorxiv.org/content/10.1101/2021.12.15.472866v1>).

Response: We thank this reviewer for pointing out. Indeed, complete Mettl3 KO is very difficult to achieve in most cells. Most published Mettl3 KO cell lines fail to achieve complete loss of METTL3 because of unpredictable alternative isoforms which resist the removal of catalytic domain, and result in substantial m6A retention (a comprehensive evaluation of these cell lines has been provided in a published literature <https://www.biorxiv.org/content/10.1101/2021.12.15.472866v1>). Mouse ESC has been reported to be the only credible cell type which could tolerate complete Mettl3 KO, and we validated the credibility of our cell line using LS-MS/MS and MeRIP-seq. We added more details about the strategy for the Mettl3 KO in the Methods section (see the schematic diagram below), which were also demonstrated in a published study (Liu, J. et al. The RNA m(6)A reader YTHDC1 silences retrotransposons and guards ES cell identity. Nature. 591, 322-326 (2021). <https://doi.org/10.1038/s41586-021-03313-9>).

2. To evaluate the soundness of the authors' data analysis, the readers would appreciate if the authors could share the raw code for the m6A detection with each tool. If it's possible, a platform (e.g. as a docker or singularity image) that could run all of the ten tools for m6A detection would greatly help the scientific community. Moreover, the authors should disclose the code of the data analysis for readers (e.g. at github).

Response: We thank this reviewer for the helpful suggestion. In our original analysis, we compared the performance of ONT tools with their default parameters. In the revised work, we have replaced these point estimates with continuous evaluation metrics including Receiver Operating Characteristic (ROC) and Precision Recall (PR) curves (Fig. 2a-d and Supplementary Fig. 2a-b). Therefore, we applied specific parameters and even modified some raw code of tools to output the results of all detected sites. The code and corresponding description are now available at https://github.com/zhongzhd/ont_m6a_detection. As this reviewer pointed out, a platform that could run all tools would greatly help the scientific community. However, we found most tools require setting up a specific working "environment". We speculate that some

research group (including us) could construct an integrated platform and address this issue in the future.

3. Since MeRIP-seq didn't have single nucleotide resolution, how the sites obtained by nanopore could be validated by MeRIP-seq in Fig. 2?

Response: We thank this reviewer for this question. The main validation datasets were generated by miCLIP and miCLIP2, which detect m6A sites at single base resolution. When using MeRIP-seq data, we roughly considered the sites falling in the peaks as true positives. We think this process is reasonable because multiple m6A sites tend to cluster as MeRIP-seq peaks (Linder et al., *Nat Methods*. **12**, 767–772, <https://doi.org/10.1038/nmeth.3453>). Though it may overestimate the precision and recall of ONT methods, we speculate that the ranking of performance would not change (compared to that using miCLIP and miCLIP2 as validation sets) (Fig. 2b, d and Supplementary Fig. 2a-b).

4. The authors need to provide adequate details for the rationale of the candidate pool of RRACH sites (line228), and how these sites are selected as the candidate pool? Even though calibrating m6A sites with KO samples could improve the accuracy, the difference is not significant for most of the tools. Could the authors show the difference between the true-positive and false-positive sites? For example, the motifs around the sites, nucleotide ratio. It might help to explain the causes of the bias.

Response: We thank this reviewer for this suggestion. We have added more details in the revised main text and Methods section. The "candidate pool" includes all the RRACH sites covered by minimum 5 reads (20 reads for m6Anet, and motifs limited to AGACT and GGACH for MINES). Then, we applied each single-mode tool to test whether these sites are m6A or not. Strikingly, a substantial number of m6A sites (from 983 to 30,207) were identified in the KO sample by most tools (Fig. 3c), suggesting a systematic overestimation of m6A abundance. Sites detected in the KO sample did not show clear enrichment around the stop codons when compared to the background (Supplementary Fig. 9c), thus implying a pervasiveness of false-positives in the results. We also randomly sampled identical number of sites (number of m6A sites detected in KO samples) from the candidate pool as random dataset for each tool, and then calculated the intersection of WT&KO and WT&Random. We found that the recurrence of m6A sites in WT samples were significantly higher in KO samples than random datasets (see Methods) (Fig. 3c). We inspected the sites detected in both WT and KO samples and found that single nucleotide polymorphisms (SNPs) and Insertions/Deletions (indels) near RRACH motifs were possible causes of this bias, especially for ELIGOS which detects m6A by comparing "errors" to the background model and treat SNPs/indels as "errors" during m6A detection (Supplementary Fig. 9d-e). In addition, homopolymer near the RRACH motif was also one of the causes of the bias (Supplementary Fig. 9d-e), as they may lead to deletion "errors" signal and inaccurate events assignment. Despite this, a proportion of sites detected in both WT and KO samples could not be explained (Supplementary Fig. 9d-e).

5. It's known that GGACT, GGACA, GGACC, and AGACT are more likely to be

methylated. The minor current differentials between methylated and unmethylated As in other motifs may be the result of their low methylation levels by nature, while not the sensitivity of the current differentials. A direct RNA nanopore sequencing of the in-vitro synthesized methylated or unmethylated oligos containing these RNA motifs would provide a clean experiment to test this hypothesis.

Response: We thank this reviewer for this question. We agree that the synthesized methylated or unmethylated oligos containing these RNA motifs would provide conclusive evidence. However, currently available datasets are generated using *in vitro* transcription with all the As replaced with m6As. This kind of artificial sequences are quite distinct from native RNA and we think the dataset is not a good reference. Alternatively, we compared the current intensity between fully methylated and non-methylated sites determined by m6A-REF-seq (see Methods). Indeed, there is no gold standard dataset representing the single-base m6A level of natural transcriptome, yet m6A-REF-seq provides a compromise to relatively quantify m6A at certain motifs.

6. As m6A-REF-seq only quantified a subset of m6A sites (RRACA sites), miCLIP sites should be used as it covers much more m6A sites in single nucleotide resolution and the peak enrichment of miCLIP sites reflects the methylation levels of the miCLIP sites.

Response: We thank this reviewer for this suggestion. We calculated the enrichment score ($FPKM_{IP}/FPKM_{input}$) to quantify the methylation levels of m6A sites which located in peaks (see Method). Unfortunately, the miCLIP datasets we used in this study did not include input data, thus we only used the enrichment score derived from MeRIP datasets in our following analysis.

Reviewer #3 (Remarks to the Author):

Zhong et. al. presents a study on comparison m6A identification of different bioinformatic tools from direct RNA sequencing of Oxford Nanopore Technology with the m6A sites identified by antibody-NGS based technique of MeRIP-seq. The comparisons were further extended to compare the m6A sites with m6A-REF-seq, miCLIP and miCLIP2 method from independent studies. The comparisons were conducted with the transcriptome of the well-known biological case of WT vs Mettl3-knockout to study m6A in mESCs. The study has a good objective to evaluate the performance of the bioinformatic tools to identify m6A from native RNA sequences at signal levels or errors levels. However, there are many points that need to be addressed in the manuscript to make fair comparison.

Response: We appreciate this reviewer's positive comments. According to the constructive suggestions, we addressed the issues point-by-point and updated the results. Now we believe this validity and rigidity have been substantially improved.

Major Comments.

- Benchmarking needs the ground truth to reference. The authors used the high confident m6A sites identified from MeRIP-seq to compare. I don't think, this is the ground truth even though the authors published their improvement on calling m6A using NGS in recent published paper (<https://www.nature.com/articles/s41592-021-01280-7.pdf>). So "benchmarking" should not be used, because we all know that there is certain level of fault

positive remains in the identified m6A sites. I strongly recommend the authors use “comparison” instead of “benchmarking”.

Response: We entirely agree that none of the current NGS method could be gold standard. Therefore, we adopted multiple independent validation sets to evaluate these tools separately, and we believe that the results would be more convincing if we obtained consistent results using different validation sets. We appreciate this comment and substitute "benchmarking" with "comparison" in the title.

- Among NGS-base methods, the m6A sites under constraints of RRACH motif identified from MeRIP-seq is consistent with m6A-REF-seq or miCLIP or not? This is a very important information that need to firstly clarify. The author showed some indirect comparison at the end (sup. Fig. 7). It is important to show first comparison result among NGS-based method and present a rationale selection of m6A sites under constraints of RRACH motif as the set to compare with DRS-based methods. From the sup. fig7, the number of identified m6A sites are approximately 18,000, 13,000 and 2,000 for MeRIP-seq, or miCLIP and m6A-REF-seq, respectively. I can see that there are several disagreements among NGS-based methods! How we can ensure the results from MeRIP-seq, which are peaks not single nucleotides is gold standard?? Should the author use combination with evident ranking (how many time the identified m6A sites present in the three methods) or intersection of all m6A sites derived from MeRIP-seq, or miCLIP and m6A-REF-seq?? This comment is very important point that need to be clearly presented at first. The small number of m6A sites identified by m6A-REF-seq resulted from low sequencing depth??

Response: We thank this reviewer for this insightful comment. Currently, there is no widely recognized gold standard method for m6A detection, and it have been reported that different NGS-based methods captured different sets of m6A sites (<https://www.nature.com/articles/s41467-019-13561-z>). We believed that the results would be more convincing if a tool performed similar when using different validation sets. Among NGS-based methods, miCLIP and miCLIP2 detect m6A sites at single-base resolution. We thus used their results as the main validation sets. When using MeRIP-seq data, we roughly considered the sites falling in the peaks as true positives. We think this process is reasonable because multiple m6A sites tend to cluster as MeRIP-seq peaks (Linder et al., *Nat Methods*. **12**, 767–772, <https://doi.org/10.1038/nmeth.3453>). Though it may overestimate the precision and recall of ONT methods, we speculate that the ranking of performance would not change (Fig. 2b, d and Supplementary Fig. 2a-b). Actually, the m6A sites detected by MeRIP-seq, miCLIP and miCLIP2 exhibited considerable overlap (see the figure below). As the reviewer suggested, we also added new results with the union or intersection of NGS methods as validation sets, which would help to avoid the influence of false negatives and false positives. Despite of the use of different validation sets, nearly the same rankings were observed (Fig. 2a-d and Supplementary Fig. 2a-b). Because of the motif limitation, m6A-REF-seq only detects m6A with ACA motifs. That is why only ~2,000 m6A sites were detected by m6A-REF-seq. However, we used the m6A sites identified by m6A-REF-seq in the quantification analysis (Fig. 5b), rather than in the tools comparison analysis as a validation set.

Figure legend: The number of m6A sites identified by MeRIP-seq, miCLIP, miCLIP2 and their intersections. The values separated by slash represent the number of sites and peaks respectively. For example, 4721 / 2673 meant 4721 sites located in 2673 peaks.

- Sequencing depth is always the important factors in calling m6A sites in either NGS-based or DRS-based methods. With the various level of expression of different transcripts, I doubt that ~1.3 million reads of generated from DRS would be sufficient to make comparison especially in low expressed transcripts? The author also reported the small fraction of reads are full length in sup. fig. 1b due to the strong 3' bias of DRS. To have a fair comparison between sequencing methods, the author should have the similar amount of data for NGS and DRS to begin with. We know that the NGS-Based methods have much better resolution because we amplified and sequenced only the DNA fragments derived from enriched region not whole transcriptome as DRS-based methods. Therefore, we need to work on this issue before performing the comparison between NGS-Based and DRS-based methods. It can be done for example by simple calculations of the sequencing depth of at all RRACH motifs then use the information to normalize the sequencing depth. The mean transcript coverage depth is not a good indicator, it is a corpse estimate. This issue recently reported in the studies of Grünberger et. al. (<https://rnajournal.cshlp.org/content/early/2021/12/14/rna.078937.121>) and Wongsurawat et. al. (<https://www.frontiersin.org/articles/10.3389/fbioe.2022.842299/full>) in bacteria and yeast system. The 3' bias of DRS is a major factor that impact the RNA modification callings therefore the authors need to make sure about sequencing depth at individual RRACH motif.

Response: We appreciate this reviewer for this constructive comment. In our original analysis we ignored the sequencing depth issue. As this reviewer suggested, we revised the related section thoroughly. Given the insufficient sequencing depth of the DRS data, a considerable number of m6A sites detected by NGS-based methods were not covered by the DRS data, leading to lower recall rates of all the tools we tested; nevertheless, the main goal of this work is focusing on ONT tools but not comparing DRS and NGS. We speculated the sequencing depth would not affect the ranking of tools. To accommodate the relatively low sequencing depth, we made two major improvements:

1) We used continuous evaluation metrics (ROC and PR curves) to compare ONT tools

on a set of sites that are covered by all tools (Fig. 2a-d and Supplementary Fig. 2a-b). For the following m6A detection using optimal cut-off, we only considered the sites covered by both NGS and DRS data for recall and precision calculation.

2) Two published DRS datasets corresponding to WT and *Mettl3* KO mESCs samples were included as replicates (Supplementary Fig. 4). In addition, we also included published DRS data from wild type Arabidopsis (Col-0) and *vir-1* mutants defective of m6A writer complex in our analysis (Supplementary Fig. 5-7).

We believe the revised results would be much more convincing and informative.

- After clear the sequencing depth issues above, cut off is the next step. Default cut-off is an intuitively recommendation by the developer that may not the optimum choice as the research filed of identification of RNA modification from native sequences/signals is still not mature yet. As the authors have a good dataset, I recommend that authors performed sensitivity analysis of the cut off and recommend to the readers. This will be a very important results for the field. For example, the authors vary cut-offs and calculate the performance matrix such as F1score (harmonic mean of precision and recall) or accuracy and the choose the best cutoff for further comparisons. These will be very useful information to the broad audiences.

Response: We thank this reviewer for this constructive comment. We also realized that default cut-off was set with variable stringency and may not be optimal for some tools. Determining the optimal cut-off according to F1 scores is a very smart strategy (Supplementary Fig. 8). In the revised work, we applied the optimal cut-offs for m6A detection analysis (Fig. 3a-b). In addition, to make a comprehensive comparison of these tools, we introduced two continuous evaluation metrics, the Area Under the Curve (AUC) of ROC and PR curves (Fig. 2a-d and Supplementary Fig. 2a-b). We believe the updated information should be much useful to broader audiences.

- The stoichiometry of m6A is another major of factor in call methylation from DRS. The tools that need reads population as input will suffer on low stoichiometry of m6A sites. On the other hand, read level identification could be handle the low stoichiometry m6A sites. Therefore, the comparison of the perform in bin of stoichiometry levels as shown in fig is good approach. However, number of m6A sites identified by m6A-REF-seq is small. The result may not reflect the overall picture. This is a limitation of this results.

Response: We thank this reviewer for this comment. We totally agree that precise m6A quantification is essential for such analysis. Unfortunately, there is no gold standard method to quantify methylation rate of every m6A site in the transcriptome. m6A-REF-seq is a compromise though it is estimated to only detect 16-25% of transcriptome-wide m6A sites in mammals. Given the common view that m6A sites with higher methylation rates produce more reads in the IP sample, we also calculated the sites enrichment scores ($FPKM_{IP}/FPKM_{input}$) from MeRIP-seq data to represent the m6A stoichiometry (Fig. 6e-f).

- Integrated analyses section as Fig7 shows the comparison of DRS vs NGS by combining the m6A sites identified by 10 tools. As the earlier results shown the low overlap between NGS-based and DRS based method, I don't think this comparison will

gain information for the reader. Authors should not do this.

Response: We thank this reviewer for this suggestion. It would be an interesting attempt to compare NGS and DRS in m6A detection, though we think the current data are not sufficient to draw any conclusion. We revised this section and refined our analysis as below:

1) We tested whether combining the results from multiple tools can improve performance. We found low overlap ratio between most pairs of tools (Fig. 7a). Integrating their results could improve the performance, suggesting that different tools may capture different sets of authentic m6A sites. Actually, we found integrated predictions achieved better performance than individual tool and using m6A sites supported by more than 4 or 5 tools achieved the highest F1 score. (Fig. 7b-c and Supplementary Fig. 13a-e).

2) We compared the m6A sites detected by DRS (merging the m6A sites of all ONT tools) to those detected by NGS (merging the m6A sites of miCLIP and miCLIP2) and inspected the sites that are captured by DRS but not by the NGS (Fig. 7d). We found that these sites showed clearer enrichment around the stop codon when supported by more ONT tools (Fig. 7f), while the overlap ratio of the m6A sites that were detected in mESCs KO samples decreased (Supplementary Fig. 13f). This implies that the NGS method failed to detect some of authentic m6A sites that were captured by the ONT tools (Fig. 7e-f and Supplementary Fig. 13f).

- For reproducibility of the result, which is the serious issue of computational biology research, the authors NEED to provide computational notebook containing, codes, command and description how all bioinformatic analyses were performed for the readers.

Response: We thank this reviewer for pointing out. We have provided such information at https://github.com/zhongzhd/ont_m6a_detection

Minor comments

Line 66 “PacBio sequencing” -> “PacBio DNA sequencing? To make sure that the detection of DNA and RNA modification is accomplished by DNA sequencing.

Response: We thank this reviewer for pointing out. PacBio has not been reported for detecting RNA modifications. We corrected this mistake in the revised manuscript.

Nanopolish need to be in the time line of the Sup. Fig. 1a. The timeline of tool should be based on publication date of the major paper of the tool.

Response: We thank this reviewer for pointing out. We revised the figure and included Nanopolish.

REVIEWER COMMENTS

Reviewer #1 (Remarks to the Author):

The authors have addressed most of the points that we have raised by including additional validation datasets, providing a more robust comparison of the m6A detection tools through the use of ROC AUC and PR AUC, as well as an analysis of the ONT-exclusive m6A sites. The revised manuscript provides a fairer comparison and therefore better overview of existing methods. The manuscript was greatly improved during the revision, with only a few remaining points to be addressed (see below).

Additional Comments

1. In the revised manuscript, Nanocompore and xPore stand out as outliers in terms of ROC AUC and PR AUC. The authors explain that this is due to these methods not returning predictions for all m6A sites, which the authors address by assigning missing predictions as false negatives. However, this procedure limits the performance of both tools when measured by the ROC AUC or PR AUC. For example Nanocompore can only achieve a maximum TPR/Recall of 0.25 and xPore can only achieve a maximum TPR/Recall of 0.4 (based on where the straight line in Figure 2c starts). As a result, the procedure by the authors enforces a low overall performance, however, this reflects the evaluation set more than the performance of these methods. Indeed, both methods perform quite well on other evaluation metrics that focus on the top sites only. Therefore, the results from the compare-mode methods do not reflect the performance of these methods, but the approach that the authors took to make the methods comparable.

a. This is similar to the evaluation of single-mode tools that only use the RRACH motifs together with tools that predict m6A on all DRACH motifs, and which was addressed by using only the intersection of sites among all tools (thereby making the ROC and PR curves comparable without disadvantages to specific tools). To address this and to achieve a fair comparison for the compare-mode methods, the authors should show the ROC plot and PR plot and ROC AUC and PR AUC for the compare-mode tools when the intersection of sites is used.

b. The fact that these two methods do not return predictions for all sites is a limitation that is worth highlighting, possibly as another plot or in the text. However, this should be highlighted separately from the performance evaluation since readers will otherwise wrongly interpret these results as inaccurate predictions when these results instead show that these methods only return predictions on a subset of sites.

c. ROC lines are comparable even when different evaluation sets are used, so the authors could also estimate the ROC AUC without adding all non-predicted sites as false negatives to these 2 methods. However, this will not work for the Precision-Recall plots, so only the evaluation of the intersection of sites from all tools can address this

2. The authors consider any ONT sites that fall within the peaks of MeRIP-seq to be true positives, but this might also reduce the performance of the ONT-based tools since such peaks might contain multiple RRACH sites (as evidenced by the difference in the ROC AUC of MeRIP-seq sites vs ROC AUC of miCLIP/miCLIP-2 sites), and only a small portion of them might contain m6A sites. Have the authors

looked at how the performance of the models will change if one aggregates the sites within each peak together? For example, by taking the maximum of the predicted probability within each peak?

3. Integrative analysis: the plot comparing the F-score of the methods will naturally show a very low F-score for sites supported by a lot of tools since there are only a handful of them. This might still suggest that those sites have very high precision (as indicated by the metagene plot) and therefore, showing the F1 plot alone might mislead the readers by suggesting that sites with a lot of support have “poor performance” as measured by a low F1 score. The authors provide such information in Supplementary Figure 13a,c but these are not mentioned in the text. To address this, the authors should show the precision and recall plot in the main figure (in addition to the F1 figure) as this would be consistent with the evaluation of the other methods and show a clearer and more transparent picture. The overall claim is consistent across all figures, but the additional analysis would provide a more comparable estimate to the individual methods in terms of precision and recall.

4. Integrative analysis: the quality of the predictions made by each model that is supported by $\geq N$ tools can differ greatly, as shown by the authors. It will be great if the authors can provide an analysis or recommendation on which of the n tools should be run if users want to have strong support for their predicted sites.

Minor comments

* Line 452: ELIGIS_diff -> ELIGOS_diff

* It is not clear why nanom6A and ELIGOS were singled out to represent current-based and error-based tools in Fig. 4c,d. The trend shown in the figures has been shown in Fig. 4a,b, and the absolute precision and recall numbers are already in the supplementary table 6

* In Fig. 4, the RRACH bar in Fig. 4 feels a bit out of place since they are located in the middle of all the other motifs

* Figure 2b,d, e and g, the x-axes labels are incorrectly aligned

* Line 506 “inadequate” not clear in this context, since there is no good way of defining an adequate balance of precision and recall. I would suggest reformulating this.

* Line 28 “largely improve performance” is exaggerated (see comment above) I would suggest to just remove “largely” here.

Reviewer #2 (Remarks to the Author):

We first appreciate the authors’ revision and feedback on our concerns for the Manuscript by Zhong et al. As we wrote in the initial comment, the authors tried to cover 10 existing tools for m6A detection from DRS nanopore data and compared the performance difference among them. While we appreciate the authors’ effort, the overall contribution of the work to the field may not be evident enough in technological advance nor the biological insights. Thus, it may not be suitable for the broad readership at Nature Communications, and would be better to be published in a journal with more focused interest.

Regardless, we provided our comments on the authors' feedback on our initial comments, hoping to be of help for the manuscript improvement.

1. We thank the authors for their disclose on the strategy for the Mettl3 KO. We are not sure about the authors' comments that "Mouse ESC has been reported to be the only credible cell type which could tolerate complete Mettl3 KO". For example, Pratanwanich et al (xPore, 2021, <https://doi.org/10.1038/s41587-021-00949-w>) published WT and METTL3-KO HEK293T DRS dataset, was this KO not a complete Mettl3 KO?
2. We appreciate the code deposit by the authors at github, and agree that the distinct setting environment for each tool may be tricky. Thus, it highlighted the importance of the disclose of Docker file or at least the authors' disclosure on the detailed working environment (i.e. the version control for different packages) for each tools which would be helpful for the community to reproduce their comparisons.
3. Since the authors agreed that MeRIP-seq doesn't have single nucleotide resolution and they wrote that "When using MeRIP-seq data, we roughly considered the sites falling in the peaks as true positives", the authors need to make sure this part is clearly written in the manuscript and that these "true positives" are roughly predicted but not experimentally confirmed. We should keep in mind that the "true positives" may not be the real true positives when interpreting the data.
4. The more disclose of method details by the authors helps the readers' understanding of their work. We appreciate the point that "single nucleotide polymorphisms (SNPs) and Insertions/Deletions (indels) near RRACH motifs were possible causes of this bias". These details highlight the important need of the DRS experiment on the in-vitro synthesized methylated or unmethylated oligos containing these RNA motifs which was initially proposed in point 5.
5. The in-vitro synthesized methylated or unmethylated oligos may provide the ground truth on the current difference between m6A vs A situation for DRS nanopore sequencing, there are initial work in the field that provided such data (e.g. Figure 6 in Workman et al 2019 <https://doi.org/10.1038/s41592-019-0617-2>)
6. We agree that combining the single-nucleotide resolution of miCLIP m6A sites with the enrichment score derived from MeRIP datasets may reflect the methylation levels of the miCLIP sites. But using the same cell type data for miCLIP and MeRIP is necessary.

Reviewer #3 (Remarks to the Author):

Sequencing depth is the main issue that produce low performance (low F1scores across all tools) when compared the results of the bioinformatics tools derived from DRS to NGS- based results.

The authors still have not raised the issues of sequencing depth that is much higher in NGS- based data than DRS data. For me, different sequencing depth between NGS and DRS will make the results incomparable. For me the conclusion of the paper is DRS for m6A identification is not the good approach, NGS-based on enriched sample (by specific antibody) is much better. If we down sampling

NGS to have a low sequencing depth like DRS, could we identify the same sites of high sequencing depth? Of course, the answer is NOT. The authors have not completely worked on this concern as I commented before.

Some bioinformatic tools were developed based in DRS synthetic data (in vitro transcription), which has high sequencing depth. The tools have very good accuracy of ROC analysis. This strongly indicate the important of sequencing depth that is a limitation of DRS data generation.

The computational simulated data of DRS with vary sequencing depths is the best way to evaluate the performance of different tools on identification of m6A.

Again the different ins sequencing depth between NGS and DRS make them incomparable.

Point-by-point response to reviewer comments

We are very grateful for the feedback from the reviewers and have revised the manuscript based on their suggestions. Our responses to the specific points raised are highlighted in orange, and the modified text in the manuscript is also marked.

Reviewer #1 (Remarks to the Author):

The authors have addressed most of the points that we have raised by including additional validation datasets, providing a more robust comparison of the m6A detection tools through the use of ROC AUC and PR AUC, as well as an analysis of the ONT-exclusive m6A sites. The revised manuscript provides a fairer comparison and therefore better overview of existing methods. The manuscript was greatly improved during the revision, with only a few remaining points to be addressed (see below).

Response: Thank you for the feedback. We have made changes to the manuscript to address the concerns raised by the reviewer. We hope that the revised manuscript provides a more robust and fair comparison of the tools. We appreciate the reviewer's support and believe that the revised manuscript adequately addresses the remaining concerns.

Additional Comments

1. In the revised manuscript, Nanocompore and xPore stand out as outliers in terms of ROC AUC and PR AUC. The authors explain that this is due to these methods not returning predictions for all m6A sites, which the authors address by assigning missing predictions as false negatives. However, this procedure limits the performance of both tools when measured by the ROC AUC or PR AUC. For example Nanocompore can only achieve a maximum TPR/Recall of 0.25 and xPore can only achieve a maximum TPR/Recall of 0.4 (based on where the straight line in Figure 2c starts). As a result, the procedure by the authors enforces a low overall performance, however, this reflects the evaluation set more than the performance of these methods. Indeed, both methods perform quite well on other evaluation metrics that focus on the top sites only. Therefore, the results from the compare-mode methods do not reflect the performance of these methods, but the approach that the authors took to make the methods comparable.

a. This is similar to the evaluation of single-mode tools that only use the RRACH motifs together with tools that predict m6A on all DRACH motifs, and which was addressed by using only the intersection of sites among all tools (thereby making the ROC and PR curves comparable without disadvantages to specific tools). To address this and to achieve a fair comparison for the compare-mode methods, the authors should show the ROC plot and PR plot and ROC AUC and PR AUC for the compare-mode tools when the intersection of sites is used.

b. The fact that these two methods do not return predictions for all sites is a limitation that is worth highlighting, possibly as another plot or in the text. However, this should be highlighted separately from the performance evaluation since readers will otherwise wrongly interpret these results as inaccurate predictions when these results instead show that these methods only return predictions on a subset of sites.

c. ROC lines are comparable even when different evaluation sets are used, so the authors could also estimate the ROC AUC without adding all non-predicted sites as false negatives to these 2 methods. However, this will not work for the Precision-Recall plots, so only the evaluation of the intersection of sites from all tools can address this.

Response: We appreciate a lot for this suggestion. Nanocompore clusters the reads into two groups based on their current features for each site, and calculates the significance of the difference between the WT and KO samples. However, these tools return "nan" for sites whose reads cannot be clustered. In addition, sites with incorrect current distributions and with a wrong methylation level prediction (KO > WT) are recommended to be filtered. As a result, these tools only return the predictions for a small fraction of sites. Thanks to this reviewer for bringing this to our attention, we assigned the lowest scores to that missing information in our previous analysis. However, as the reviewer pointed out, this process unfairly labeled sites with missing information as false negatives. We apologize for the mistake and have now updated our results (ROC plot, PR plot, ROC AUC and PR AUC) for the compare-mode tools using the intersection of sites as recommended (Fig. 2c-d and Supplementary Fig. 2-7). We have also highlighted the fact that these two tools omit a large fraction of sites in the precision/recall/F1 score section. We believe these changes have greatly improved the manuscript and provided a fairer comparison and better overview of existing methods.

2. The authors consider any ONT sites that fall within the peaks of MeRIP-seq to be true positives, but this might also reduce the performance of the ONT-based tools since such peaks might contain multiple RRACH sites (as evidenced by the difference in the ROC AUC of MeRIP-seq sites vs ROC AUC of miCLIP/miCLIP-2 sites), and only a small portion of them might contain m6A sites. Have the authors looked at how the performance of the models will change if one aggregates the sites within each peak together? For example, by taking the maximum of the predicted probability within each peak?

Response: We appreciate this suggestion and agree with the importance of considering real-life scenarios when evaluating performance. We observed that the ROC AUC of MeRIP-seq was lower than that of miCLIP/miCLIP-2, although the ranking of performance was largely unchanged. We tried aggregating the sites within each peak together, but were concerned about the accuracy of this approach given that multiple m6A sites tend to cluster in MeRIP-seq peaks according to Linder et al. (Nature Methods, <https://doi.org/10.1038/nmeth.3453>). As suggested by the reviewer, we implemented a strategy of selecting the maximum predicted probability within each peak, which resulted in a significant improvement of ROC AUC (as shown in the provided figure below). Notably, the ranking of performance for the tools remained largely unchanged.

Supplementary figure for response. a: any ONT sites that fall within the peaks of MeRIP-seq are considered to be true positives. b: ONT sites with maximum of the predicted probability within each peak are selected to represent the results for each peak.

3. Integrative analysis: the plot comparing the F-score of the methods will naturally show a very low F-score for sites supported by a lot of tools since there are only a handful of them. This might still suggest that those sites have very high precision (as indicated by the metagene plot) and therefore, showing the F1 plot alone might mislead the readers by suggesting that sites with a lot of support have “poor performance” as measured by a low F1 score. The authors provide such information in Supplementary Figure 13a,c but these are not mentioned in the text. To address this, the authors should show the precision and recall plot in the main figure (in addition to the F1 figure) as this would be consistent with the evaluation of the other methods and show a clearer and more transparent picture. The overall claim is consistent across all figures, but the additional analysis would provide a more comparable estimate to the individual methods in terms of precision and recall.

Response: Thank you for your suggestion. We have moved Supplementary Figure 13a and 13c to the main figure in order to provide a clearer and more transparent comparison. We appreciate your feedback and hope that this improves the clarity of the manuscript.

4. Integrative analysis: the quality of the predictions made by each model that is supported by $\geq N$ tools can differ greatly, as shown by the authors. It will be great if the authors can provide an analysis or recommendation on which of the n tools should be run if users want to have strong support for their predicted sites.

Response: We appreciate this suggestion and agree that it can be helpful to explore the best combination of different tools for users with specific needs. While it can be challenging to define “custom needs” without bias, we believe that our analysis can provide useful guidance for users to select suitable tools or combinations based on their specific requirements. For example, users may want to consider using m6Anet or xPore if they are looking for strong support for their predicted m6A sites (as shown in Fig. 2a-d, Fig 3a-b and Fig. 7c). If users are interested in obtaining a large number of sites, merging the results from m6Anet and xPore could be a good option, as these tools have relatively low overlap

(as shown in Fig. 7a). In addition, users who are focused on maximizing recall rate may want to include the results from DiffErr, DRUMMER or ELIGOS_diff, which use different detection strategies than m6Anet and xPore but also have high accuracy (as shown in Fig. 2a-d, Fig 3a-b and Fig. 7a, c). We believe that this information will be helpful for those who are interested in identifying the most appropriate tool for their specific needs.

Minor comments

* Line 452: ELIGIS_diff -> ELIGOS_diff

Response: Thank you for pointing out. We have corrected this word.

* It is not clear why nanom6A and ELIGOS were singled out to represent current-based and error-based tools in Fig. 4c,d. The trend shown in the figures has been shown in Fig. 4a,b, and the absolute precision and recall numbers are already in the supplementary table 6

Response: Thank you for your suggestion. We have removed Fig. 4c-d as they appear to be somewhat redundant. We appreciate the reviewer's feedback and hope that the revised manuscript is now clearer and more concise.

* In Fig. 4, the RRACH bar in Fig. 4 feels a bit out of place since they are located in the middle of all the other motifs

Response: Thank you for pointing out. We have removed Fig. 4c-d.

* Figure 2b,d, e and g, the x-axes labels are incorrectly aligned

Response: We thank this reviewer for pointing out. We adjusted the x-axis labels of Figure 2b, d, e and g.

* Line 506 "inadequate" not clear in this context, since there is no good way of defining an adequate balance of precision and recall. I would suggest reformulating this.

Response: We appreciate the suggestion made by this reviewer. In revising the text, we have rephrased it as follows: "it was challenging to balance precision and recall rates for some tools".

* Line 28 "largely improve performance" is exaggerated (see comment above) I would suggest to just remove "largely" here.

Response: We thank this reviewer for this suggestion. We have made this edit to the text.

Reviewer #2 (Remarks to the Author):

We first appreciate the authors' revision and feedback on our concerns for the Manuscript by Zhong et al. As we wrote in the initial comment, the authors tried to cover 10 existing tools for m6A detection from DRS nanopore data and compared the performance difference among them. While we appreciate the authors' effort, the overall contribution of the work to the field may not be evident enough in technological advance nor the biological insights. Thus, it may not be suitable for the broad readership at Nature Communications,

and would be better to be published in a journal with more focused interest. Regardless, we provided our comments on the authors' feedback on our initial comments, hoping to be of help for the manuscript improvement.

Response: We appreciate these helpful comments from the reviewer. We have addressed the remaining concerns and carefully revised the manuscript. Undoubtedly, ONT DRS has demonstrated superiority over NGS-based methods due to its single-molecular resolution and ability to generate full-length reads, as well as its capacity to simultaneously detect multi-layers of RNA processing events. In recent years, a number of tools have been developed to detect RNA modifications from ONT DRS data, but little is known about the capabilities and limitations of these tools. Here we showed that this comprehensive comparison between these tools provides a guide for best practices in m6A detection and quantification, which may serve as a foundation for future research. We believe that our work is timely and essential for the following reasons:

- 1) Our findings suggest that m6A detection by ONT DRS is practical but requires continuous optimization. Sites predicted as m6A by DRS tools can be verified by NGS-based methods and have abundant biological features relevant to specific species.
- 2) Using a m6A-deficient negative control, such as complete Mettl3 KO samples or IVT, can improve the performance of these tools. SNP, indel and homopolymer adjacent to RRACH motifs are important causes of false-positive sites and should be removed from the predicted results when using single-mode.
- 3) Detection capabilities varied among motifs, with m6A sites in AGACT/GGACH showing better evaluation metrics than GAACH, and the trend is consistent among all tools. This is because only minor current differentials occur between modified and unmodified A's in some motifs, leading to large amounts of background variation that can make A and m6A indistinguishable. The quality of the quantitative information obtained by current-based computational tools varied among motifs, with acceptable results observed only in some motifs.
- 4) Integrating the predictions from multiple tools can improve performance, as using m6A sites supported by more than 4 or 5 tools achieved the highest F1 score. ONT DRS methods and NGS-based methods are complementary in that some apparently authentic m6A candidate sites can only be detected by DRS or by NGS.

In summary, we believe that our work will draw more attention from researchers and lead to the development of more effective tools in this field. As a promising but still immature new field, ONT DRS stands to benefit greatly from this study.

1. We thank the authors for their disclose on the strategy for the Mettl3 KO. We are not sure about the authors' comments that "Mouse ESC has been reported to be the only credible cell type which could tolerate complete Mettl3 KO". For example, Pratanwanich et al (xPore, 2021, <https://doi.org/10.1038/s41587-021-00949-w>) published WT and METTL3-KO HEK293T DRS dataset, was this KO not a complete Mettl3 KO?

Response: Thank you for this question. Upon further review, we realized that our previous response stating that "mouse ESC has been reported to be the only credible cell type which

could tolerate complete *Mettl3* KO” is not accurate. According to a recent study by Poh et al. (<https://pubmed.ncbi.nlm.nih.gov/35853000/>), certain cell lines other than mESCs are able to survive after *Mettl3* knockout. However, the cell line used in xPore (Pratanwanich et al., Nature Biotechnology 2021) showed evident m6A residual after *Mettl3* KO (as shown in Fig. S8 in the paper, see below). In our own laboratory, we have also made great efforts to knockout *Mettl3* in widely used cell lines such as HEK293T and Hela, but have reached the same conclusion that most cell lines are resistant to *Mettl3* KO in practice (Poh et al., PLoS Biol., 2022). We apologize for any confusion caused by our previous statement.

(from Xpore, Pratanwanich et al., Nature Biotechnology 2021)

2. We appreciate the code deposit by the authors at github, and agree that the distinct setting environment for each tool may be tricky. Thus, it highlighted the importance of the disclose of Docker file or at least the authors' disclosure on the detailed working environment (i.e. the version control for different packages) for each tools which would be helpful for the community to reproduce their comparisons.

Response: We appreciate these positive comments and constructive advice from the reviewer. We have provided more detailed information about the working environment for each tool on our github page (https://github.com/zhongzhd/ont_m6a_detection). Thank you for your feedback.

3. Since the authors agreed that MeRIP-seq doesn't have single nucleotide resolution and they wrote that "When using MeRIP-seq data, we roughly considered the sites falling in the peaks as true positives", the authors need to make sure this part is clearly written in the manuscript and that these "true positives" are roughly predicted but not experimentally confirmed. We should keep in mind that the "true positives" may not be the real true positives when interpreting the data.

Response: We thank the reviewer for bringing this to our attention. We also recognized that the MeRIP-seq data used as the evaluation set is a compromise and should be clearly stated in the manuscript. We also implemented a new strategy in which we selected the highest-scoring site rather than all sites falling in the same peaks as the true positives (as shown in the supplementary figure below). This analysis resulted in a significant improvement of ROC AUC (as shown in the figure below). However, the ranking of the tools' performance largely remained unchanged. We apologize for any confusion caused by our initial submission and appreciate the opportunity to clarify our methodology.

Supplementary figure for response. a: any ONT sites that fall within the peaks of MeRIP-seq are considered to be true positives. b: ONT sites with maximum of the predicted probability within each peak are selected to represent the results for each peak.

4. The more disclose of method details by the authors helps the readers' understanding of their work. We appreciate the point that "single nucleotide polymorphisms (SNPs) and Insertions/Deletions (indels) near RRACH motifs were possible causes of this bias". These details highlight the important need of the DRS experiment on the in-vitro synthesized methylated or unmethylated oligos containing these RNA motifs which was initially proposed in point 5.

Response: We appreciate these positive comments from the reviewer. We fully agree that in-vitro synthesized methylated or unmethylated oligos would be the gold standard for model training, although generating such a dataset including all possible motifs remains a big challenge. We have added this discussion in the revised manuscript. Thank you for your feedback.

5. The in-vitro synthesized methylated or unmethylated oligos may provide the ground truth on the current difference between m6A vs A situation for DRS nanopore sequencing, there are initial work in the field that provided such data (e.g. Figure 6 in Workman et al 2019 <https://doi.org/10.1038/s41592-019-0617-2>)

Response: We thank the reviewer for this suggestion. We fully agreed that in-vitro synthesized oligos with methylated or unmethylated A in determined location, as demonstrated by Workman et al. 2019, provide a ground truth for testing the current difference between m6A and A. In their work, they show that the current intensity of GGACU decreases after methylation (as shown below). However, generating such a dataset including all possible motifs remains a big challenge. We speculate that research groups, including ours, will use this strategy to assess the current difference between m6A and A in other motifs in the near future. Thank you for your feedback.

(from <https://doi.org/10.1038/s41592-019-0617-2>)

6. We agree that combining the single-nucleotide resolution of miCLIP m6A sites with the enrichment score derived from MeRIP datasets may reflect the methylation levels of the miCLIP sites. But using the same cell type data for miCLIP and MeRIP is necessary.

Response: We appreciate these positive comments from the reviewer. We would like to clarify that the miCLIP and MeRIP data are from the same cell type (mESCs). Thank you for your feedback.

Reviewer #3 (Remarks to the Author):

Sequencing depth is the main issue that produce low performance (low F1scores across all tools) when compared the results of the bioinformatics tools derived from DRS to NGS-based results.

The authors still have not raised the issues of sequencing depth that is much higher in NGS- based data than DRS data. For me, different sequencing depth between NGS and DRS will make the results incomparable. For me the conclusion of the paper is DRS for m6A identification is not the good approach, NGS-based on enriched sample (by specific antibody) is much better. If we down sampling NGS to have a low sequencing depth like DRS, could we identify the same sites of high sequencing depth? Of course, the answer is NOT. The authors have not completely worked on this concern as I commented before.

Some bioinformatic tools were developed based in DRS synthetic data (in vitro transcription), which has high sequencing depth. The tools have very good accuracy of ROC analysis. This strongly indicate the important of sequencing depth that is a limitation of DRS data generation.

The computational simulated data of DRS with vary sequencing depths is the best way to evaluate the performance of different tools on identification of m6A.

Again the different ins sequencing depth between NGS and DRS make them incomparable.

Response: Thank you for this constructive comment. We apologized for any confusion regarding the comparison of DRS with NGS in our work. Our goal is actually to compare the performance of different ONT DRS tools in m6A detection using validation sets from orthogonal NGS methods. To make this comparison more objective, we introduced

continuous evaluation metrics such as ROC/PR in the section comparing the performance of ten computational tools; in the section on the intrinsic bias of ONT tools in m6A detection, we used precision/recall/F1 score under optimal cut-off. To minimize the influence of sequencing depth, we required a minimum coverage of 5 or 20 DRS reads for this evaluation (as shown in Supplementary Table 3). This means that the m6A sites/peaks detected by NGS methods (MeRIP, miCLIP and miCLIP2) but not covered by ONT reads was NOT included in the validation sets.

We agree that relatively low sequencing depth can compromise the evaluation metrics such as F1 score due to the loss of true positive sites, especially for sites with low methylation rate (for example, at least 10 DRS reads are required to identify a site with a 0.1 methylation rate). In fact, as more DRS reads are sampled, more m6A sites with low methylation rate would be determined. In the revised results, we extended our analysis to candidate sites covered by at least 20 or 50 DRS reads separately (Supplementary Fig. 9 and corresponding source data). We found that increasing the coverage from 5 to 20 resulted in a better F1 score for all tools, with most of them achieving a better precision and some achieving a better recall. However, increasing the coverage from 20 to 50 did not result in better performance for most tools (and even slightly worse for some tools). Importantly, the ranking of tools was nearly the same regardless of different coverage requirements. We discussed this issue further in the section “Detection capability is influenced by sequence coverage and m6A stoichiometry”.

We also agree that the values of evaluation metrics (precision, recall, and F1 score) in our paper may not accurately reflect the real performance of these ONT tools due to the low sequencing depth of the DRS data, which could potentially underestimate their performance. Notably, we have found that the performance of most ONT tools is significantly enhanced when applied to Arabidopsis, which generates much higher sequence coverage. We have detailed this in a section: “Detection capability is influenced by sequence coverage and m6A stoichiometry”. We have also highlighted this point in the revised version of discussion section. However, as these ONT tools were applied to the same DRS dataset, we believed that these evaluation metrics are comparable between tools, even if the absolute values of evaluation metrics may not reflect their true performance.

We apologize for any confusion caused by our initial submission. We believe that ONT DRS has significant potential for m6A detection and that careful assessment and continuous optimization of current tools will be key in the process of replacing NGS-based methods.

REVIEWERS' COMMENTS

Reviewer #1 (Remarks to the Author):

Overall the authors have satisfied our concerns with the manuscript. The authors have modified the comparisons that involve xPore and Nanocompare by evaluating their performances in shared positions while highlighting the difference in the number of sites predicted by the two methods is highlighted in the precision/recall/F1 score section. Furthermore, the authors have also moved the precision and recall plot from the Supplementary Figure to the Main Figure of the section “Integrated analyses of multiple tools”, which we believe provides a more transparent and balanced comparison between all the methods and users will be able to choose accordingly given the trade-offs. We believe that, given the availability of many m6A detection tools, this work is timely and will be a fine addition to the collection of m6A detection literature. This work also provides a useful benchmark for users to follow and other methods developers to build their work on.

We only have one small concern regarding the authors’ response regarding the comparison with MeRIP-seq labels. While we do acknowledge the validity of the authors' concern regarding multiple m6A sites clustering together in MeRIP-seq peaks, validating the performance of the methods under the assumption that all RRACH sites within each peak to be methylated will greatly depress the performance of each model. This is evidenced by the improvement of the ROC AUC of all the models. While the overall ranking is not changed, it is perhaps better for the authors to acknowledge this in the main text and provide the figure in the response as a supplementary figure.

Reviewer #2 (Remarks to the Author):

We appreciate that the authors took time to write back to us on our comments. As scientists, we hope that all research efforts could be rewarded well as publications to the research community. The current manuscript can be published somewhere. Whether the journal is Nature Communications, it is upon the editor's decision.

Reviewer #3 (Remarks to the Author):

The authors improved the manuscript significantly and mentioned the critical issue of sequencing depth in the manuscript. I recommend the author point the sequencing depth issue of DRS in the abstract. This is the important for the reader to get at first.

REVIEWERS' COMMENTS

Reviewer #1 (Remarks to the Author):

Overall the authors have satisfied our concerns with the manuscript. The authors have modified the comparisons that involve xPore and Nanocompore by evaluating their performances in shared positions while highlighting the difference in the number of sites predicted by the two methods is highlighted in the precision/recall/F1 score section. Furthermore, the authors have also moved the precision and recall plot from the Supplementary Figure to the Main Figure of the section "Integrated analyses of multiple tools", which we believe provides a more transparent and balanced comparison between all the methods and users will be able to choose accordingly given the trade-offs. We believe that, given the availability of many m6A detection tools, this work is timely and will be a fine addition to the collection of m6A detection literature. This work also provides a useful benchmark for users to follow and other methods developers to build their work on.

Response: Thank you for the feedback. We appreciate the reviewer's many useful suggestions and positive comments, which have significantly enhanced the quality of our work.

We only have one small concern regarding the authors' response regarding the comparison with MeRIP-seq labels. While we do acknowledge the validity of the authors' concern regarding multiple m6A sites clustering together in MeRIP-seq peaks, validating the performance of the methods under the assumption that all RRACH sites within each peak to be methylated will greatly depress the performance of each model. This is evidenced by the improvement of the ROC AUC of all the models. While the overall ranking is not changed, it is perhaps better for the authors to acknowledge this in the main text and provide the figure in the response as a supplementary figure.

Response: We appreciate the reviewer's suggestion. We have incorporated this additional information into the main text and have included a figure in the supplementary information (Supplementary Fig. 2c).

Reviewer #2 (Remarks to the Author):

We appreciate that the authors took time to write back to us on our comments. As scientists, we hope that all research efforts could be rewarded well as publications to the research community. The current manuscript can be published somewhere. Whether the journal is Nature Communications, it is upon the editor's decision.

Response: We are grateful to the reviewer for their thoughtful comments and suggestions.

Reviewer #3 (Remarks to the Author):

The authors improved the manuscript significantly and mentioned the critical issue of sequencing depth in the manuscript. I recommend the author point the sequencing depth issue of DRS in the abstract. This is the important for the reader to get at first.

Response: We appreciate the helpful comments provided by the reviewer. We have now included the suggested addition in the abstract.